# Integration of multi-level dental diversity links macro-evolutionary patterns to ecological strategies across sharks

Roland Zimm[1]*, Vitória Tobias Santos[2], Nicolas Goudemand[1]

[1]ENS de Lyon, Institut de Génomique Fonctionnelle de Lyon, Lyon, France;
[2]Laboratoire de Biologie du Developpement de Villefranche-sur-Mer (LBDV, UMR 7009), Institut de la Mer de Villefranche (IMEV), Sorbonne Université, CNRS, Villefranche-sur-Mer, France

**\*For correspondence:**
zimm.roland@gmail.com

**Competing interest:** The authors declare that no competing interests exist.

**Abstract** The exceptional dental diversity in sharks is frequently used as a proxy for ecological function. However, functional inferences from morphology need to consider morphological features across different organizational scales and spatial resolutions. Here, we compare morphological features ranging from sub-dental patterns to whole dentitions within a large ensemble of species encompassing all extant shark orders. Although taxa scoring high for different heterodonty measures are distributed throughout the phylogeny, the two shark superorders show a different degree of modularity between mono- and dignathic heterodonty as well as substantial differences in ecological niches. Intriguingly, we observe two alternative ways of increasing dental complexity: either at the tooth- or dentition-level. Correlating heterodonty and single-tooth complexity with ecological and life-history traits, we find that pelagic and demersal species evolve dental complexity in different ways. We track trait variability as a function of genetic distance, thus quantifying dental trait adaptability at different resolutions. Overall, intermediate resolution levels, namely the degree of monognathic heterodonty, predict ecological traits best but carry a relatively low phylogenetic signal, suggesting a more dynamic adaptability on shorter evolutionary timescales. This raises macro-evolutionary interpretations about the evolvability of nested modular phenotypic structures, with important implications for paleo-ecological inferences from sequentially homologous traits.

## Editor's evaluation

This important study combines morphological and genetic information of teeth and dentitions of extant sharks with a novel morphometric tool combination and phylogenetic information to better understand the relevance of hierarchical organization of functional traits to identify trait adaptability. Indicating developmental and ontological nestedness, the overall results are convincing and have the potential to be fundamental to the application of dental traits for paleoenvironmental and palaeoecological reconstruction in sharks, vertebrates in general.

## Introduction

Teeth have been used as a powerful proxy for ecological function and adaptive evolution across vertebrates (*Cooper et al., 2023*; *Jernvall et al., 1996*; *Fischer et al., 2022*; *Eronen et al., 2010*; *Segall et al., 2023*; *Evans and Sanson, 2003*). As a hyper-diverse structure displaying considerable morphological variation within meso-evolutionary frameworks (*Streelman and Albertson, 2006*; *Lafuma et al., 2015*; *Davis et al., 2016*), tooth shape tends to be fine-tuned for food acquisition and mastication strategies (*Evans and Sanson, 2003*). This is particularly important for the reconstruction and

**eLife digest** Teeth are among the most diverse and complex organs in the animal kingdom that have evolved over millions of years to accommodate a wide range of diets and habitats. Their shapes vary broadly between and within species, and even among individuals, which often possess multiple tooth types within a single dentition.

Mammals and sharks both exhibit particularly high tooth diversity, having diverged from a common ancestor several hundred million years ago. Comparing their dentitions, therefore, provides a valuable opportunity to investigate how these organs evolved, how they develop, and what they reveal about past ecosystems and animal behaviors.

Although mammalian and shark teeth form through distinct developmental mechanisms, both groups share key principles. For example, in both groups, large-scale features – such as tooth position within the mouth – are established before finer details, such as small cusps or serrations.

Zimm et al. sought to determine which levels of anatomical organization are most informative for understanding how composed functional traits like dentitions evolve and adapt. This question is critical because many biological functions arise from the integration of traits across multiple spatial scales – from fine serrations and cusps to differences between teeth and jaws – and establishing a methodological framework for comparing these levels has remained a major challenge. Moreover, while isolated shark teeth have been studied previously, little is known about how sharks have adapted to different ecological niches.

To address this, Zimm et al. developed a new methodology for comparing dentitions using an online collection of shark teeth combined with genomic data. Their analyses revealed strong statistical differences in types of dental complexity between the two major shark groups and the different environments they inhabit, especially the deep sea and the open ocean. For example, the way neighboring teeth differ from each other is more shaped by their ecological niches rather than by genetic relationships, while the latter explains more details within teeth or dental differences between jaws.

These findings provide new insights into how integrated biological systems adapt and evolve. Because shark dentitions are tightly linked to ecological function, this work may also offer new avenues for studying long-term changes in marine ecosystems. Together, these perspectives could play an important role in developing both general and specific predictions about how the biosphere may evolve in a rapidly changing world.

study of past ecosystems, since fossil teeth are often abundant, providing critical information about ecological niche occupancy (*Jernvall et al., 1996*; *Eronen et al., 2010*; *Fulwood et al., 2021*; *Evans, 2013*). While complexity and shape of isolated teeth are functionally informative (*Whitenack et al., 2011*; *Whitenack and Motta, 2010*; *Frazzetta, 1988*; *Ballell and Ferrón, 2021*), teeth tend to work as a whole (or partial) dentition, forming an emergent functional unit. Thus, single-tooth morphology is only one of several different organizational levels - from sub-dental features (e.g. serrations) to entire dentitions - that matter for specific functional aspects and their integration. This amounts to a limitation for paleontological studies often relying on isolated teeth whose relative positions are deduced indirectly (*Shimada, 2005*). The functional integration of teeth, as whole dentitions or by regional subfunctionalization, is further illustrated by the widespread occurrence of heterodonty (juxtaposition of differently shaped teeth). Like mammals, sharks exhibit conspicuous tooth morphological variation at different scales, from single tooth to dentition levels. Interestingly, this is not a recently evolved feature, as heterodont arrangements of multicuspid teeth are described for Devonian sharks (*Maisey et al., 2014*). Deep similarities across sharks and mammals (*Debiais-Thibaud et al., 2015*; *Thiery et al., 2022*) point towards a kernel of developmental mechanisms capable of generating highly diverse dental morphologies both between and within individuals, which has been explored experimentally and computationally (*Salazar-Ciudad and Jernvall, 2010*; *Zimm et al., 2023a*; *Harjunmaa et al., 2014*).

While a system of tooth classes is well-established in mammals (*Mitsiadis and Smith, 2006*; *Sharpe, 2000*), systematic knowledge about heterodonty biases in sharks is sparse. For specific low-rank taxonomic groups, patterns of tooth shape and size variation along the jaw are often diagnostic (*Shimada, 2002*; *Türtscher et al., 2022*). Some of these patterns have been linked to feeding

mechanics, emphasizing the importance of dentition-level perspectives when connecting morphology and ecological functions (*Tapanila et al., 2020*), while stark tooth morphology differences exist between sexes and age cohorts (*Berio et al., 2020*; *Türtscher et al., 2022*; *French et al., 2017*; *Cullen and Marshall, 2019*; *Purdy and Francis, 2007*). This is why dentitions of several species have been dissected morphometrically within proximate phylogenetic contexts (*Berio et al., 2020*; *Türtscher et al., 2021*; *Cullen and Marshall, 2019*). Complementarily, many studies have used isolated teeth across wider taxonomic levels, identifying clade- and ecotype-specific clusters and distributions (*Cullen and Marshall, 2019*; *Bazzi et al., 2021a*; *Bazzi et al., 2018*, *Ballell and Ferrón, 2021*; *Goodman et al., 2022*). Representing even higher organizational levels, jaw geometry, cranial shape, musculature, and other anatomic macro-features have been linked to feeding strategies (*López-Romero et al., 2023*; *Moss, 1977*; *Wilga et al., 2007*), underlining the adaptive interplay of traits on different scales. This suggests that a class-wide analysis of functional associations between dental variation and complexity at different scales and ecological functions might be highly informative. Here, we elucidate such macro-patterns in the light of environmental and life-history traits, applying a novel morphometrics tool combination. Specifically, we expect that complex dental traits on different organizational levels, from fine denticles to differences between jaws, exhibit specific contributions to functional adaptation in an either universal or clade-specific manner. This analysis may critically contribute to understanding the specific relevance of hierarchical organization of functional traits throughout an entire vertebrate class, a significant question across ecology, paleontology, and evolution.

## Results

### Heterodonty is widespread across sharks

Heterodonty reflects the degree of tooth shape variation within a given individual. Such variation is not always subtle and gradual, limiting the use of common homology-based morphometrics tools (e.g. landmark-based approaches), arguably biasing efforts towards species with gradual heterodonty (*Türtscher et al., 2022*; *Berio et al., 2020*) and against class-level comparisons (*Bazzi et al., 2018*). Thus, we aspire to account for the complexity of heterodonty and quantify it across the diversity of extant sharks. We calculate different types of within-toothrow heterodonty, namely differences between neighboring teeth within the same jaw (sequential monognathic heterodonty, HMS), between all teeth within the same jaw (total monognathic heterodonty, HMT), and between teeth from the same relative positions in opposing jaws (dignathic heterodonty, HDG), *Figure 1*. By quantifying different, but rigorously defined, types of heterodonties, we reduce complex, multidimensional features to one-dimensional measures and are able to compare morphologically heterogeneous taxa. Since heterodonty measures quantify shape variation between units, they can be considered a proxy for high-level (i.e. jaw-level) complexity. We also devise different proxies for single-tooth morphological complexity based on 2D-outline characteristics (see *Figure 1—figure supplement 2*), enabling us to contrast tooth and dentition-level features.

In order to assess the prevalence of heterodonty across sharks, we collected 2D shape information about complete or nearly complete dentitions from 51 species, using an open data collection (J-elasmo: http://naka.na.coocan.jp/), representing all extant shark orders. The above-defined measures identified substantial levels of heterodonty within all major clades of sharks, albeit to varying degrees (*Figure 2*). While no significant difference emerges between the two superorders Squalomorphii (squalean sharks) and Galeomorphii (galean sharks) regarding total and maximal monognathic heterodonty ($P_{HMT}$ = 0.3386, $P_{HMTmax}$ = 0.928, two-sided Wilcoxon test), sequential heterodonty is lower ($P_{HMS}$ = 0.0755), and dignathic heterodonty is significantly higher ($P_{HDG}$ = 0.0058) in Squalomorphii. Overall, these findings do not generally support strong superorder-level biases that might facilitate, or canalize, dental variation within individuals.

### Heterodonties are correlated

It is conceivable that some species may show a high degree of between-jaws tooth variation without exhibiting high variation between adjacent teeth, and vice versa. Therefore, we need to elucidate the relationship between different heterodonty measures. In sum, we find strong correlations between monognathic and dignathic heterodonties (HMT~HMS: $R$=0.9197, HDG~HMT: $R$=0.4236, HDG~HMS: $R$=0.3062), suggesting that variation within and between jaws is not independent (*Figure 1B*); the

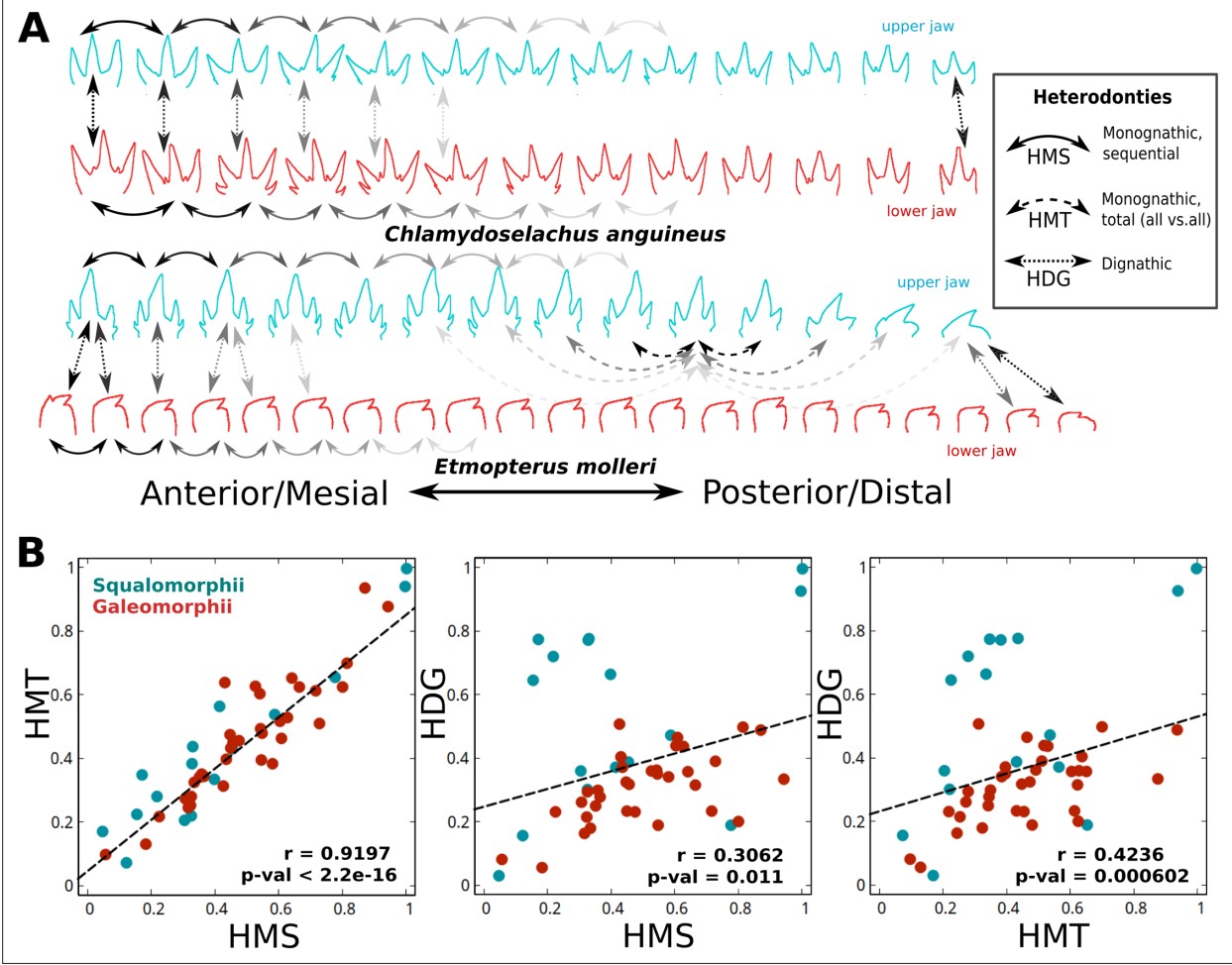

**Figure 1.** Overview of heterodonty measures. (**A**) Heterodonty, a dentition-level disparity measure, can refer to (1) average differences between successive teeth along the jaw: sequential monognathic heterodonty: HMS, (2) averaged differences among all teeth within the same jaw, total heterodonty: HMT, (3) differences between pairs of teeth belonging to opposing jaws, also termed dignathic heterodonty: HDG. Here, these different measures are partially illustrated using differently dashed arrows. While the upper dentition of *Chlamydoselachus anguineus* shows fairly similar teeth between upper (turquoise) and lower (red) jaw, the lower dentition of *Etmopterus molleri* displays conspicuous dignathic heterodonty. (**B**) Heterodonty measures (*Figure 1—figure supplement 1*) are correlated. Monognathic heterodonties (HMS, HMT), as well as dignathic heterodonty (HDG), are plotted against each other, revealing positive correlations. Colors encode phylogenetic clades (teal: Squalomorphii, red: Galeomorphii). Dashed lines show linear regression; p-values are based on Pearson's correlation test.

The online version of this article includes the following figure supplement(s) for figure 1:

**Figure supplement 1.** Different methods to quantify tooth morphological distance.

**Figure supplement 2.** Overview of tooth-level complexity measures.

**Figure supplement 3.** Correlations between measures.

**Figure supplement 4.** Tooth-level complexity measures are complementary.

main outliers being squalean sharks, which, on average, exhibit low monognathic but high dignathic levels of heterodonty (*Figure 2—figure supplement 2A*). The highest levels of both monognathic and dignathic heterodonty are found within Hexanchidae featuring highly specialized dentitions, whereas *Mustelus*, *Squatina*, *Nebrius,* and *Squalus* occupy the opposite end of the distribution. Strikingly, the latter genera exhibit very different tooth morphologies, ranging from plaque-like (*Mustelus mustelus*), unicuspid and smooth (*Squatina dumeril*), to asymmetrically bent (*Squalus mitsukurii*) and complex, multicuspid, teeth (*Nebrius ferrugineus*), indicating that there may be no trivial correlation between tooth-level complexity and dentition-level complexity. This is quantified by low-to-medium correlations between different complexity and heterodonty measures (HMS~Complexity: $R=0.196$, HMT~Complexity: $R=0.1275$, HDG~Complexity: $R=-0.063$). Overall, most galeomorphs show a more

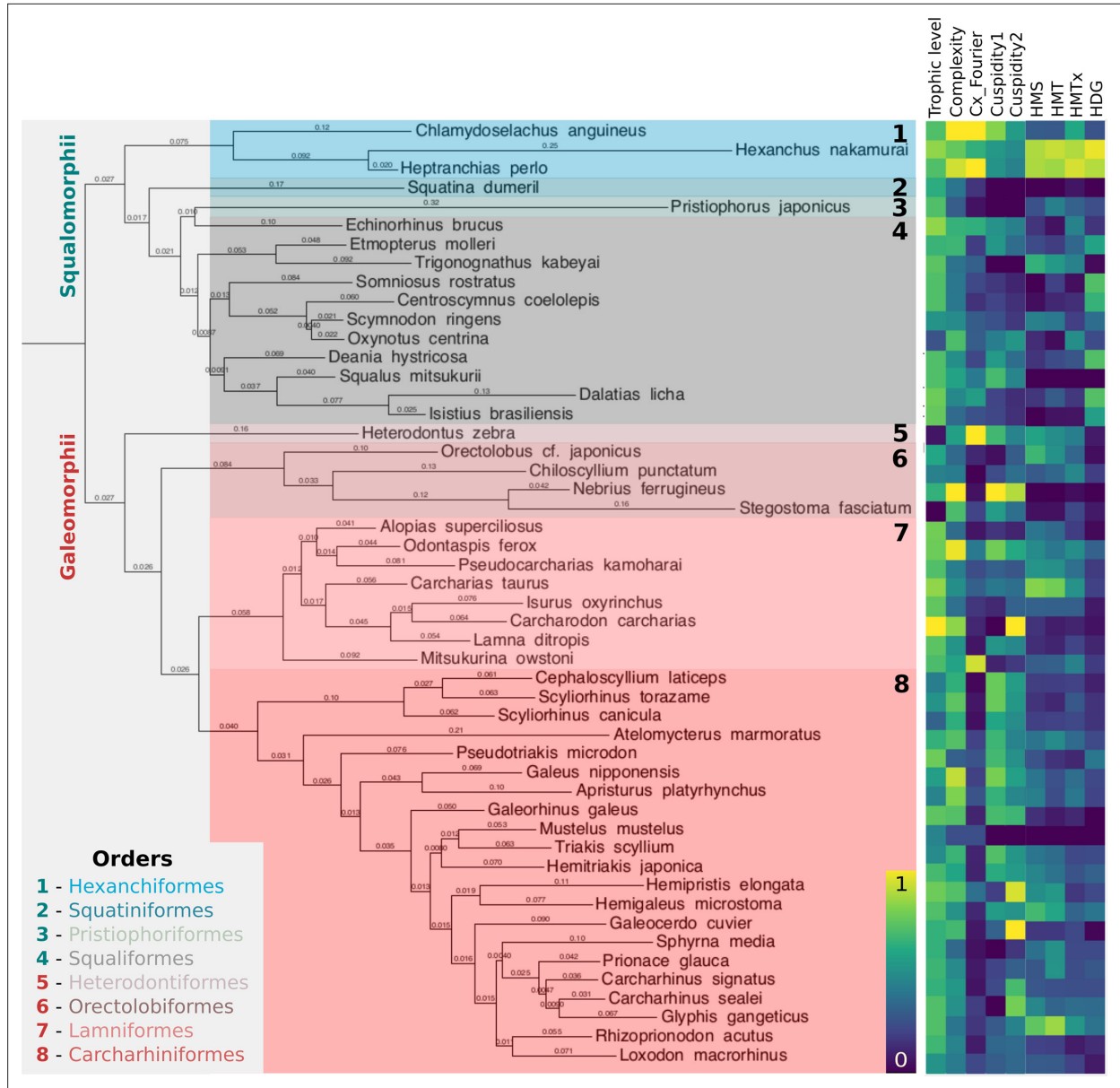

**Figure 2.** Heterodonty is widespread across all shark clades. We selected 51 species across the entire Selachimorpha, representing most of the extant shark diversity. Displayed branch lengths are proportional to genetic distance (see Materials and methods) and taxonomic orders are distinguished by background (and font) color. The adjacent heatmap shows species-wise measures of trophic level, tooth-level complexity (average of different measures, *Figure 1—figure supplement 2*), Fourier-based tooth-level complexity, cuspidity (1: coarse, 2: fine), and globally normalized heterodonty measures, *Figure 1*, *Figure 1—figure supplement 1*. HMS/HMT: sequential/total monognathic heterodonty, HDG: dignathic heterodonty, HMTx: maximal heterodonty between any two teeth of the same jaw, Cx: tooth-level complexity.

The online version of this article includes the following figure supplement(s) for figure 2:

**Figure supplement 1.** Single tooth discrete Fourier PCA.

**Figure supplement 2.** Heterodonty ratios exhibit differences across shark phylogeny.

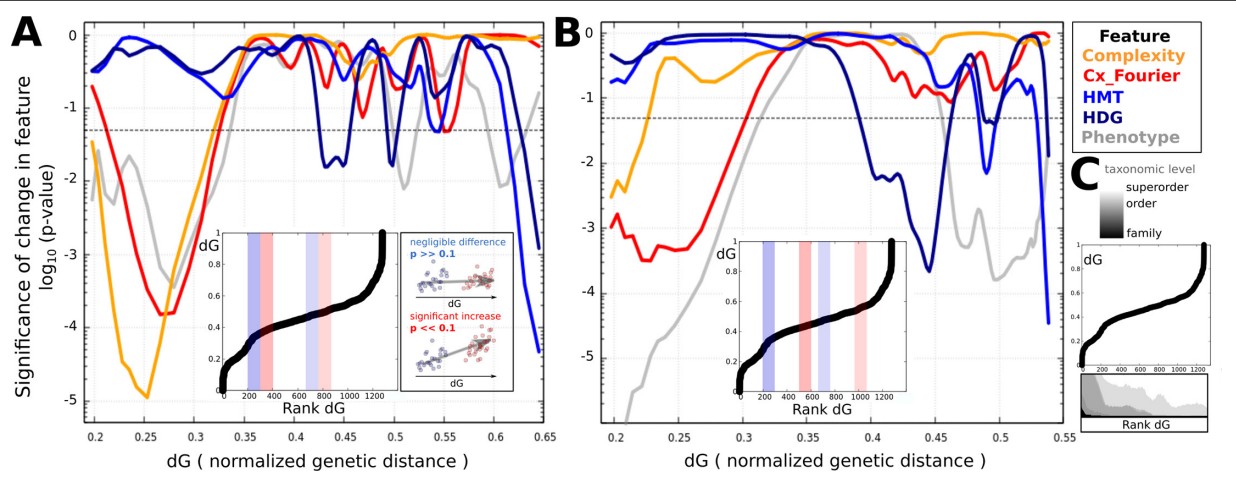

**Figure 3.** Differences in heterodonty show no significant increase with genetic distance for low- to intermediate taxonomic levels, unlike tooth-level complexity. We ordered all pairs of species by normalized genetic distance (dG) and calculated p-values (one-sided Wilcoxon test) for significance of difference of overall tooth-level complexity (orange), Fourier-based complexity (red), total monognathic heterodonty (HMT, blue), dignathic heterodonty (HDG, dark blue), and total phenotypic distance (gray), between two subsets of 100 species pairs each. Total phenotypic distance is based on position-wise tooth shape comparisons. Subsets were defined as containing the $n_{th}$ to the $n+100_{th}$ species pair ordered by dG, for sliding (incrementally increasing) n. The two subsets were (**A**) subsequent or (**B**) 200 ranks apart, in order to account for different scales of comparison. Here, the lines connecting p-values for all n are Bezier-smoothened and plotted against dG of the highest-ranked species pair within the respective lower set. A dotted line marks the 0.05-level of statistical significance. Inlets show the relationship between dG and ordered ranks and examples of two pairs of subsets (higher: red, lower: blue). For explanatory purposes, schematic examples of two pairs of sets with low (red) and high (blue) p-values are displayed beside. (**C**) For orientation, we display the taxonomic compositions of the ordered species pair sets, with black representing the portion of pairs from the same family (only a few), light gray representing pairs from the same superorder, and white pairs stemming from different superorders, with intermediate shades of gray referring to intermediate taxonomic levels. HMS: sequential monognathic heterodonty, HDG: dignathic heterodonty, Cx: tooth-level complexity.

The online version of this article includes the following figure supplement(s) for figure 3:

**Figure supplement 1.** Phenotypic distance increases steadily with genetic distance.

**Figure supplement 2.** Features show different ranges of strong and weak correlations with genetic distance.

**Figure supplement 3.** Taxon sensitivity of correlations between genetic and phenotypic distances.

**Figure supplement 4.** Phylogenetic signal.

gradual heterodonty pattern than squalomorphs, as measured by the ratio of the maximal shape difference and HMS between any two teeth (*Figure 2—figure supplement 2B*).

## Patterns of heterodonty and dental complexity across phylogenetic distances

The widespread occurrence of heterodonty and tooth-level complexity (*Figure 2*) across the entire shark phylogeny suggests repeated evolution of these features.

Thus, we are interested in analyzing how they diverge over evolutionary timescales. While monognathic heterodonty in particular appears scattered across all major clades, we find that dignathic heterodonty presents a stronger clade-specific distribution, with tooth shape differences between upper and lower jaw to be both qualitative and quantitative in Squaliformes, Charcharhinidae, Hemigaleidae, and Hexanchidae, as well as in the genera *Hemitriakis* and *Pseudotriakis*. In the case of Hexanchidae, dignathic heterodonty is not present in Mesozoic taxa (*Kriwet and Klug, 2014*), suggesting that dignathic homodonty is plesiomorphic within this clade.

In order to test how dynamically tooth complexity features vary at different phylogenetic scales, we correlate genetic distances with differences in dental morphological descriptors. This analysis reveals the absence of significant positive correlation between genetic distance and heterodonty differences for genetically close or moderately distant species (*Figure 3*, *Figure 3—figure supplement 2*). Only the genetically most distant species exhibit significantly higher heterodonty differences.

However, since the extreme tail-end of the displayed distributions is largely driven by a small number of taxa (*Figure 3—figure supplement 3*), it does not represent a generalizable pattern. This finding implies that heterodonty can change relatively unconstrainedly, suggesting substantial evolvability. Conversely, differences in tooth-level complexity increase significantly between very closely and intermediately related species. However, this does not apply to every specific measure of tooth-level complexity individually, as differences in cuspidity are uncorrelated with genetic distance (*Figure 3—figure supplement 2*). Overall, our findings suggest that there are tooth-level complexity measures such as Fourier-analysis-based complexity (Cx.four) that might serve as a moderately better proxy for relatedness than heterodonty (*Figure 3*, *Figure 3—figure supplement 4*).

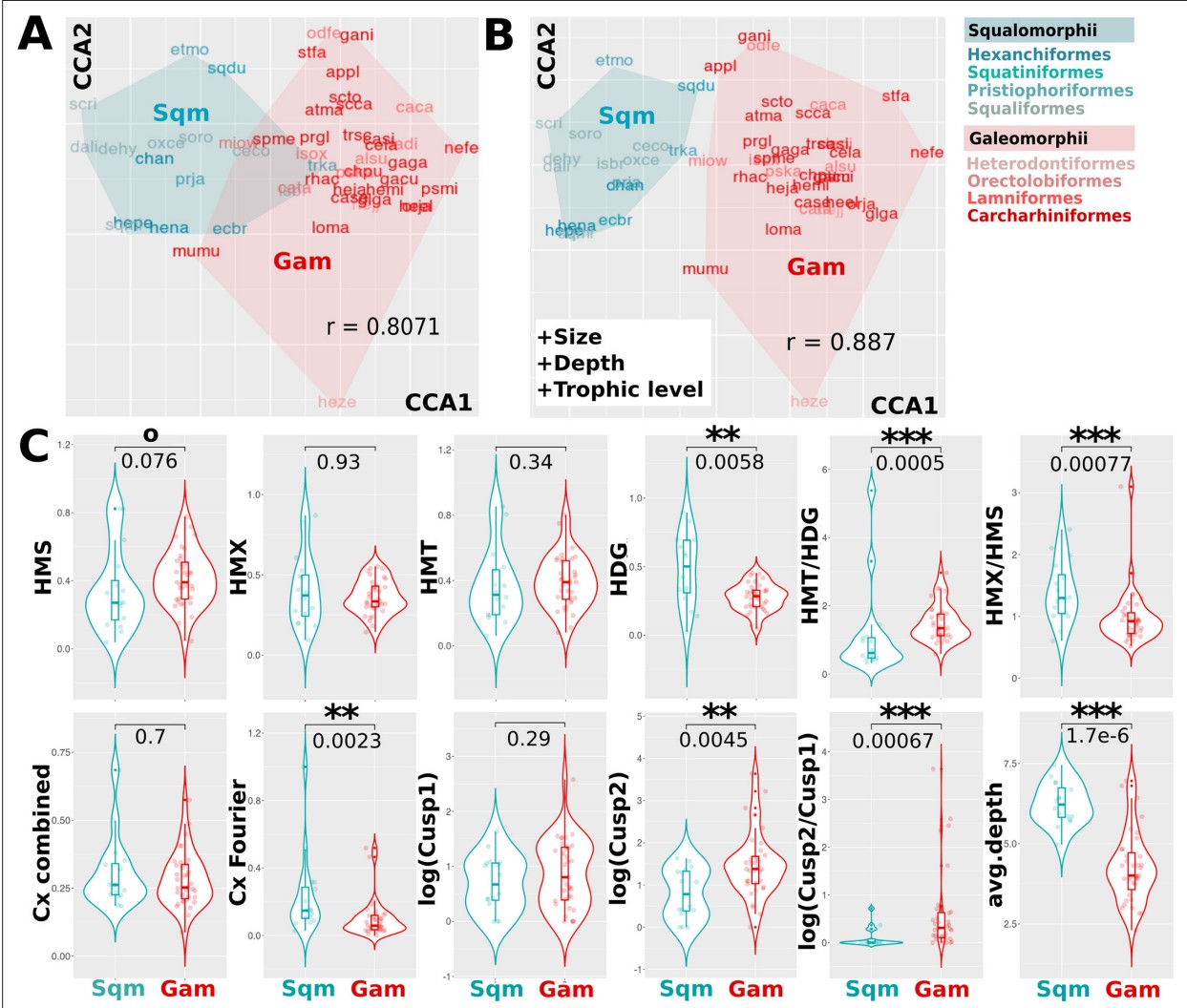

**Figure 4.** Heterodonty and tooth-level complexity measures separate shark superorders. (**A**) Canonical correlation analysis (CCA) reveals combinations of heterodonty and tooth-level complexity measures that are specific for the two superorders, squalean (Sqm, teal) and galean (Gam, red) sharks. (**B**) The two main clades are separated more clearly if ecological traits are included into the canonical analysis. The colors of the displayed species acronyms correspond to the respective orders, as displayed beside. (**C**) Violin plots contrast specific features and feature combinations in Squalomorphii and Galeomorphii, with p-values plotted above (Wilcoxon test). Monognathic and dignathic heterodonty, the ratio between the two, Fourier-based tooth-level complexity (Cx_Fourier), and heterodonty and cusp ratios, as well as depth, show significant differences between the two clades. Cx_combined is the sum of all tooth-level complexity measures, Cusp1 and Cusp2 are coarse and fine cuspidity, respectively. Boxes show quartiles and whiskers the respective adjacent values; significances: 0.1>p > 0.05: °, 0.05>p > 0.01: *, 0.01>p > 0.001: **, 0.001>p: ***. HMS/HMT: sequential/total monognathic heterodonty, HDG: dignathic heterodonty, HMX: maximal heterodonty between any two teeth of the same jaw, Cx: tooth-level complexity.

The online version of this article includes the following figure supplement(s) for figure 4:

**Figure supplement 1.** Depth and size ranges per species.

**Figure supplement 2.** Modularity of heterodonty is correlated with habitat depth.

Despite the absence of a substantial mid-range phylogenetic signal for specific heterodonty measures, we tested whether combinations of these measures differ between main shark clades. Using canonical correlation analysis (CCA), we are able to separate Squalomorphii and Galeomorphii, the two shark superorders (*Figure 4A and B*). While including only morphological features allows for separating the Squalomorphii from 80% of Galeomorphii along the first canonical axis (*Figure 4A*), adding some important ecological characteristics leads to a complete separation of the superorders (*Figure 4B*). Interestingly, the ratio between monognathic and dignathic heterodonty and graduality of heterodont change appears to be better separators than each heterodonty measure in isolation (*Figure 4C*). Consistent with the higher correlation between differences in dental complexity and genetic distance, we find that certain tooth-level complexity measures, such as fine cuspidity (and the ratio between fine and coarse cuspidity) and Fourier-based complexity, show a significant clade-specific range of values (*Figure 4C*). Finally, our study also reveals significant clade-specific differences in ecological features, especially depth, suggesting that the identified heterodonty and tooth complexity patterns may at least partly represent patterns of adaptive morphological changes (*Figure 4C*).

## Two contrasting strategies emerge

In addition to phylogenetic and ecological groups, we find that shark species diverge along two disparate directions when plotting monognathic heterodonty against Fourier complexity (*Figure 5*).

While the first group (G1) shows high Fourier tooth-level complexity but low monognathic complexity, the second one (G2) presents the reverse pattern. Teeth in G1 tend to be smoother, more obtuse, and asymmetric relative to G2. Significant differences emerge when comparing the ratios between (a) coarser and finer cusp numbers, (b) different heterodonty measures, and (c) outline-vs. angle-based tooth similarity or complexity measures (*Figure 5C*). Leveraging combinations of these measures reveals a specific shape pattern, with G1 featuring asymmetric, compact teeth that vary little within jaws, and G2 featuring more excentric or triangular teeth that are part of morphologically heterogeneous gradually changing dentitions. We also collected specific information on prey categories coarsely associated with different trophic guilds or feeding strategies. Interestingly, trophic differences between the two groups are not salient. However, species belonging to G1 tend to inhabit deeper regions, while G2 species are found closer to the surface, including both proximal (shores) and distal (open ocean) environments (*Figure 5A–B*).

## Correlations between ecological traits and heterodonty

Since tooth shapes and their arrangement within dentitions are expected to be fine-tuned towards specific niches, we evaluated correlations with ecological trait proxies (habitat, food, and body size). We used both linear correlation models and canonical variate correlations between dental measures and ecological features (*Figure 6A*, *Figure 6—figure supplement 1*).

Although partially overlapping, these two approaches yield distinct correlation profiles, owing to different methodologies. Especially monognathic heterodonty measures emerge as important diagnostic predictors in the canonical analysis paradigm. Intriguingly, we find that dental features tend to correlate more strongly with habitat than with trophic categories. Reciprocal correlation profiles separate residents of shallow and open-sea habitats, with species inhabiting the deep sea potentially presenting yet another profile of correlation with heterodonty and tooth complexity traits (*Figure 6A*, *Figure 6—figure supplement 2*). Across trophic guilds, differences emerge between nektonic (vertebrates and cephalopods) versus bottom-dwelling prey classes (crustaceans, non-cephalopod mollusks, and diverse small invertebrates, *Figure 4—figure supplement 1*). Finally, coarse cuspidity is distinctive for the difference between bottom-vs.-water column feeding strategies, actively hunted prey, and body size, while fine cuspidity is informative about size, and deep-vs.-shallow habitat distinction. Together with the previously described 'alternative strategies' (*Figure 5*), we conclude that 'demersal' sharks tend to evolve dental complexity at the single tooth level, whereas 'pelagic' sharks tend to evolve it at the dentition level. Globally, we observe distinct correlation patterns between dental descriptors and ecological traits. Intriguingly, these correlations are overall strongest for monognathic heterodonties (*Figure 6*), suggesting more general differences in their relative importance for ecological function and patterns of morphological evolution.

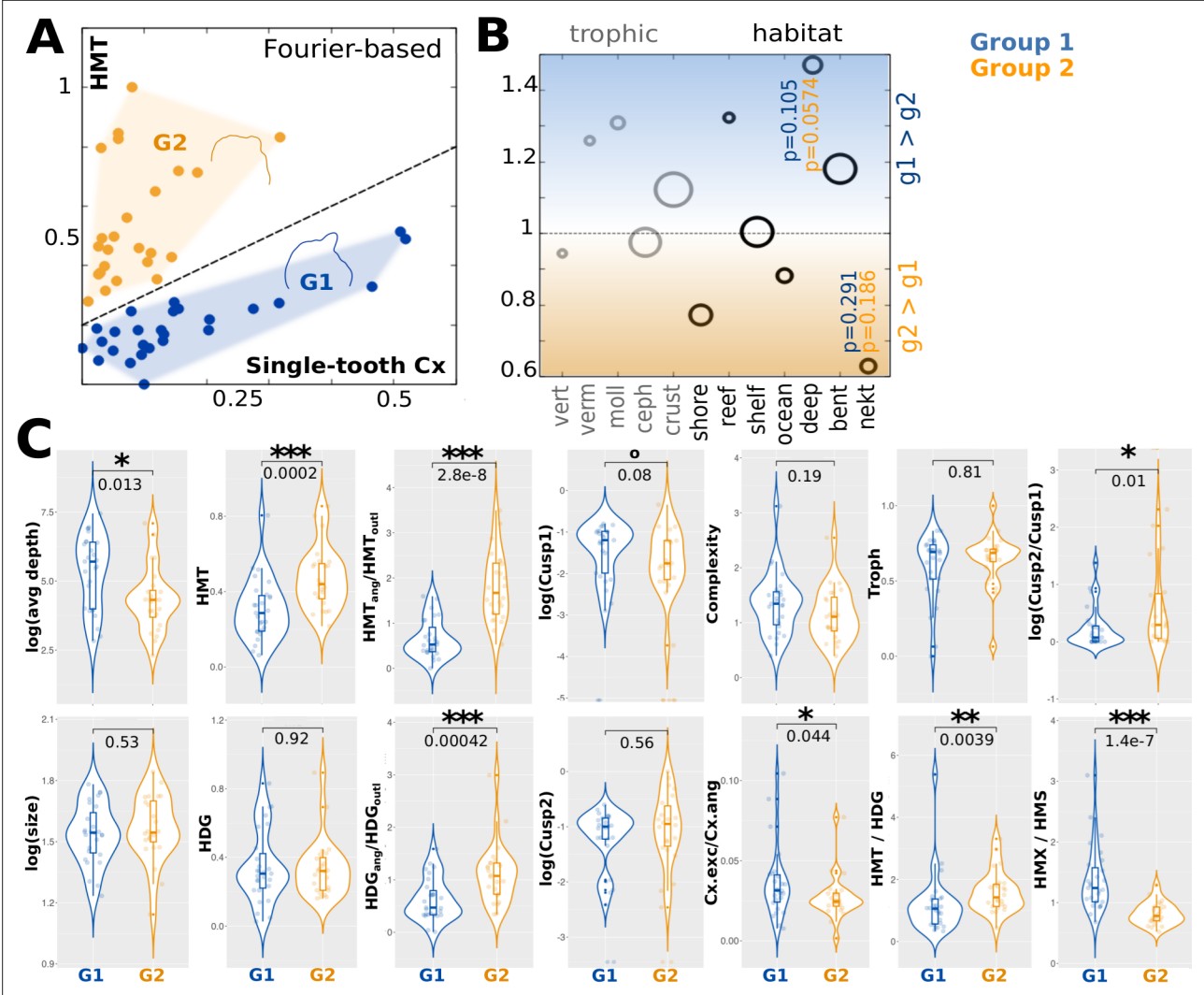

**Figure 5.** Combinations of heterodonty and tooth-level complexity measures reveal two distinct strategies. (**A**) Many shark species show, roughly, either high single-tooth complexity and low monognathic heterodonty (both Fourier-based), or vice versa, but rarely exhibit high values for both. Thus, two different clusters emerge, here denoted as group 1 (G1) and group 2 (G2). Blue marks the former (G1, n=27), orange the latter group (G2, n=24). Shown outlines display, respectively, mean tooth shapes for both groups. (**B**) Group-wise enrichments of trophic and habitat features, each ring presents the ratio between the respective percentages of species belonging to G1 divided by the ones belonging to G2 and the expected unbiased ratio. Ring size reflects the number of species per ecological category. p-values (two-sided binomial test, for G1 and G2, respectively) are annotated for the most significant differences. (**C**) Violin plots visualizing further group-specific characteristics with p-values (Wilcoxon test). Notably, the groups show divergent ratios between heterodonty measures based on outlines ($X_{outl}$) and outline angles ($X_{ang}$), corresponding to the heterodonty measures EMD, HED, and SAO vs. OAD and ADD (cf. Materials and methods). Cx.exc comprises complexity measures based on excentricity (OCR, OAR, OIR), Cx.ang (ANS, ASC, AND, OPC) measures based on angle complexity. Cx(OIR) is the minimal ratio between the areas of inscribed and escribed circles. Boxes show quartiles and whiskers the respective adjacent values; significances: $0.1 > p > 0.05$: °, $0.05 > p > 0.01$: *, $0.01 > p > 0.001$: **, $0.001 > p$: ***. HMS/HMT: sequential/total monognathic heterodonty, HDG: dignathic heterodonty, HMTx: maximal heterodonty between any two teeth of the same jaw, Cx: tooth-level complexity, Troph: mean trophic level.

## Discussion

### Heterodonty contributes to dental diversity in sharks

Although separated by over 400 Ma of evolution (*Janvier, 1996*), sharks and mammals show remarkable dental variation both between and within individuals and species. We show quantitatively that, within sharks, this diversity is not restricted to specific clades but evolves in a clade-independent way. This suggests conserved developmental mechanisms capable of producing a large range of potentially adaptive tooth shapes (*Salazar-Ciudad and Jernvall, 2010*; *Zimm et al., 2023a*). Indeed, at

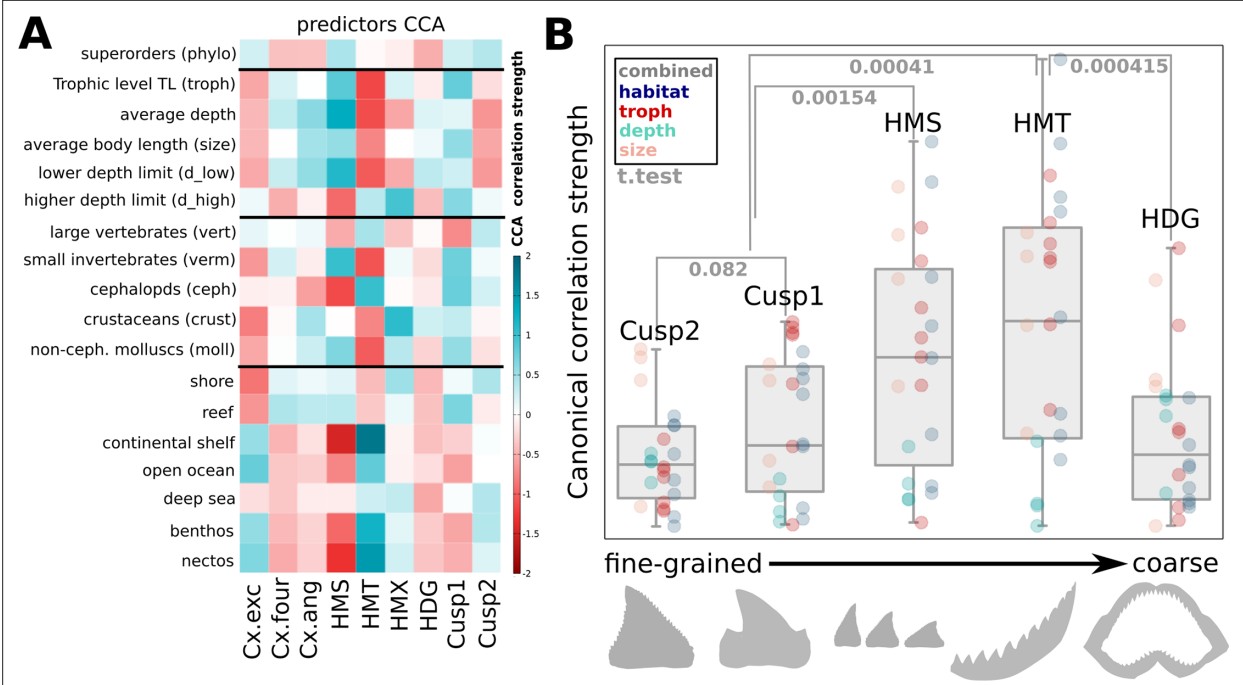

**Figure 6.** Ecological relevance of dental shape descriptors varies across resolution levels. (**A**) Different heterodonty and tooth-level complexity measures show specific correlations with ecological features, such as body size, depth, prey guilds, and habitats. Red and teal hues indicate correlation strengths of CCA1 for linear combinations of the predictors (red: negative correlations; teal: positive correlations). Cx.exec: comprises tooth-level complexity measures based on excentricity (OCR, OAR, OIR); Cx.four: Fourier-based complexity (DFS); and Cx.ang: angle-based complexity measures (ANS, ASC, AND, OPC). (**B**) Box plots summarizing canonical correlation strengths as a function of resolution/scale of the descriptors. Correlation strengths were found highest for monognathic heterodonty, while fine cuspidity yielded the lowest average correlation. As those measures represent, roughly, morphological trait differences (or complexity) on different resolution levels from fine cusps to differences between jaws, they are plotted in ascending order from the finest to coarsest scale. Canonical correlation strength can be used as a proxy for average relevance, suggesting size scale-dependent differences in ecological trait importance. Dot colors denote different types of ecological traits, while the gray boxes contain the combination of all traits, showing quartiles and extremes of the distribution. Displayed p-values were calculated using a Student's t-test. Underlying shapes are included for illustrative guidance. HMS/HMT: sequential/total monognathic heterodonty, HDG: dignathic heterodonty, HMX: maximal heterodonty between any two teeth of the same jaw, Cx: tooth-level complexity.

The online version of this article includes the following figure supplement(s) for figure 6:

**Figure supplement 1.** Linear correlations between tooth complexity descriptors and ecological traits.

**Figure supplement 2.** Similarity of canonical correlation profiles between ecological traits.

**Figure supplement 3.** Scale-dependent ecological relevance is robust against moderate data modifications.

**Figure supplement 4.** Summary: Correlations of features per resolution scale and macro-phylogeny do not reflect ecological importance.

the level of single-tooth morphologies, *in silico* approaches to mammalian and shark odontogenesis suggest a comparable capacity of generating phenotypic diversity (*Salazar-Ciudad and Marín-Riera, 2013*; *Zimm et al., 2023a*). However, at the level of dentitions, graduality differences of variation between adjacent mammalian teeth reflect the role of Hox genes in differentiating discrete tooth classes, which remains to be shown in sharks (*Mitsiadis and Smith, 2006*; *Sharpe, 2000*). Shark tooth development, on the other hand, involves the concerted activation of dental stem cells residing deeply within the dental lamina whose local differences in geometry, biomechanics, and signaling are likely to be critical for the way tooth shapes change along the jaw, too (*Fraser et al., 2020*; *Meredith Smith et al., 2018*). In contrast to the often abrupt tooth shape changes caused by differential Hox gene activation, spatial cues from the dental lamina may underlie the frequent gradual tooth shape changes seen in many sharks.

Our analysis reveals compelling differences between the two main superorders of sharks, Squalomorphii and Galeomorphii (*Figure 4*). While in the latter, the different heterodonty measures are strongly correlated (*Figure 1B*, *Figure 4—figure supplement 2*), this is less the case for squalean sharks. Together with the selective absence of dignathic, but not monognathic, heterodonty in the

stem-group hexanchoid *Notidanoides muensteri* (*Kriwet and Klug, 2014*), this finding suggests clade-specific modularity between mono- and dignathic heterodonty among Squalomorphii. We also show that Squalomorphii exhibit larger but less gradual dental variation along the jaw than Galeomorphii (*Figure 4C*). This might reflect particularly high heterodonty in Hexanchidae - functional and conserved throughout the fossil record (*Adnet, 2006*; *Kriwet and Klug, 2014*), and the cutting-blade tooth rows widespread in Squaliformes that depend functionally on interlocked homodont teeth (*Underwood et al., 2016*). In addition, Galeomorphii exhibit serrated teeth more frequently than Squalomorphii, another specific trait that can be linked to feeding strategies (*Figure 4C*).

Although gradual variation has been documented and quantified in some galean species (*Berio et al., 2020*; *Türtscher et al., 2022*; *French et al., 2017*; *Cullen and Marshall, 2019*; *Goodman et al., 2022*; *Purdy and Francis, 2007*), we are not aware of any previous study comprehensively quantifying heterodonty throughout sharks. Interestingly, jaw-level differences between the superorders may indicate different feeding strategies involving differences in dental characteristics and jaw shape (*López-Romero et al., 2023*; *Cooper et al., 2023*). Different ecological strategies between the superorders are, from a statistical perspective, supported by our finding that combining morphological and ecological descriptors allowed for a clearer separation of the galean and squalean sharks (*Figure 4B*), suggesting a link between ecology, heterodonty, and phylogeny. Strong functional, adaptive pressures may also underlie increased phenotypic evolution rates in Squalomorphii (*López-Romero et al., 2023*). While a single-tooth-based analysis across sharks did not clearly separate morphospace regions for orders and superorders, morphospace occupancy was biased (*Bazzi et al., 2021a*).

Taken together, we find that most significant dental trait differences between squalean and galean sharks are in line with high levels of functional specialization. Yet, whether developmental or purely functional constraints ultimately underlie these differences remains to be elucidated. Answering this important question may require the quantification of heterodonty in fossil dentitions which would allow calibrating the pace of phenotypic change within lineages. However, complete dentitions are rarely preserved in the fossil record, leaving a comprehensive analysis to future researchers.

## Heterodonty predicts ecological features

Emerging correlations between ecological traits and heterodonty indicate a functional value of quantitative tooth shape variation. Specific correlations, however, may vary by method (one-to-one versus canonical correlation), highlighting that dental adaptations to ecological niches are best described by a non-trivial combination of features. The finding of stronger correlations with habitat-related versus trophic traits might *prima facie* appear contradictory. Yet, many sharks are opportunistic feeders; completeness of trophic information varies between species, and prey composition may change seasonally and between age cohorts; for example (*Baremore et al., 2008*; *Vögler et al., 2008*).

Habitats correlate with feeding habits and can serve as a coarser, yet more inclusive, proxy for a species' primary food sources. Intriguingly, we found striking similarities in heterodonty correlation patterns between specific habitat and trophic characteristics, suggesting discrete clusters that represent major ecological-functional strategies: (1) shallow-water habitats, crustacean/small invertebrate diets, (2) open-sea habitats, large preys, with a sub-cluster of deep-sea cephalopod feeders. It is tempting to interpret these clusters in terms of different feeding mechanics (*Cooper et al., 2023*). Many shark species specialized in hunting larger vertebrates benefit mechanically from dental serrations, absence of larger cusps, and low-to-intermediate heterodonty (*Smith et al., 2015*; *Whitenack et al., 2011*; *Whitenack and Motta, 2010*; *Frazzetta, 1988*). Biomechanical studies showed that serrated teeth reduce tear and shear stresses, while impeding puncturing, reducing their suitability for smaller or hard-shelled prey (*Whitenack et al., 2011*; *Whitenack and Motta, 2010*; *Frazzetta, 1988*). Interestingly, several squalean taxa exhibit dignathic heterodonty with interlocking asymmetric lower teeth and simpler arrow-shaped upper teeth, which cooperate in a grasping-sawing mechanism (*Cooper et al., 2023*). Consequently, increased diagnostic heterodonty may have enabled the emergence of mechanistically complex feeding strategies particularly across Squalomorphii (*Underwood et al., 2016*; *Moss, 1977*). It is tempting to speculate that this mechanism co-emerged with an adaptation to deep-water habitats within Squalomorphii, enabled by higher modularity of mono- and dignathic heterodonty in this group. Deep-water species within Carchariniformes (deep-sea catsharks), on the other hand, do neither show increased dignathic heterodonty nor large differences to shallow-water relatives, which may be due to clade-specific constraints. Many benthic feeders, including many

deep-sea and reef dwellers, employ complex collecting-crushing or ambushing-grasping strategies aimed at smaller, often hard-shelled, prey, in line with diversified tooth shapes and assemblies along the jaw. In contrast, many smaller reef-inhabiting species catch small free-swimming animals using multi-cuspid dentitions (*Moss, 1977*; *Wilga et al., 2007*). Complementarily, feeding strategies often involve jaw-cartilage/skeletal-level adaptations, indirectly affecting tooth numbers and heterodonty gradients (*Moss, 1977*; *Sadier et al., 2023*). Together, habitat-heterodonty associations reflect how different environments and prey guilds underlie the dynamic evolution of a finite set of functional feeding strategies with specific signatures both at the tooth and dentition level. We have found strong correlations between combinations of phenotypic features and ecological traits, particularly habitat depth. Given the prominence of dental features for ecometric reconstructions of paleo-climates and mostly terrestrial ecosystems (*Eronen et al., 2010*), it is becoming clear that shark dental features can be highly diagnostic for marine habitats (e.g. depth), and that integrating information from several levels of organization, where available, might enable even more precise ecosystem reconstructions.

## Different levels of complexity characterize distinct ecological strategies

We have identified two morphological trends that are distinguished by an inverse relationship between tooth-level complexity and monognathic heterodonty (dentition-level complexity). This implies that species tend to either increase tooth-level or dentition-level complexity, but rarely both of them. Feeding strategies of group one, associated with deep-sea and benthic habitats, involve increased collecting and crushing of hard-shelled prey animals, leveraging specialized individual tooth shapes and complexity. Such specializations often involve adaptations of the entire feeding apparatus, i.e. modifications to jaw cartilage shape, articulation, and musculature, allowing for suction-based food acquisition mechanisms (*Moss, 1977*; *Wilga et al., 2007*), with possible implications for dental shape. Conversely, the second group uses more homodont, high-cuspid/serrated dentitions to catch and dismember swimming prey. Such specializations require concerted fine-tuning on several levels, thereby strongly selecting against intermediate morphologies. At the single-tooth level, teeth adapted for specialized diets often occupy extreme positions within morphospaces (*Bazzi et al., 2021b*), in line with this assumption. Thus, the discreteness of the two trends, visualized by within-morphospace divergence, suggests morphological discreteness of highly specialized functional mechanisms.

## The importance of multi-scale modular complexity

Our analysis shows that diversification of functionality in a composed morphology featuring repetitive structures involves changes in complexity across scales. This is a common observation among 'serially homologous' traits (*Bateson, 1892*; *Bateson, 1894*; *Wagner, 1989*), whose units share developmental mechanisms that, during evolution, may accumulate divergent features ultimately leading to individualization (*Sémon et al., 2025*). This is exemplified by limbs, vertebrae, and ectodermal appendages, which include dentitions. Interestingly, such organs often nest repeated sub-structures at multiple levels, for example feather branches within feathers, or digits on limbs (*Zimm et al., 2023b*; *Prum and Brush, 2002*; *Young et al., 2011*). Within dentitions, teeth are arranged in regular rows but feature themselves repeated sub-structures, namely cusps and smaller cusplets, making them a prominent example of serially organized structures (*Stock, 2001*; *Luo et al., 2007*; *Roth, 1984*). Differences between units often stem from local differences in developmental regulation within the tissue background in higher-level structures (*Jernvall and Thesleff, 2000*; *Tucker et al., 1998*). This is in line with rather gradual patterns of dental variation in specific shark clades (*Türtscher et al., 2022*; *Berio et al., 2020*; *Bazzi et al., 2018*; *Shimada, 2002*), while only a few species, for example Hexanchidae, show conspicuously discrete shape transitions. By comparing complexity across several nested levels of dental organization, this study is an important extension from the common focus on a single level of morphological organization.

With respect to function and ecology, we see that monognathic heterodonties have a significantly stronger correlation with ecological traits than dignathic heterodonty and cuspidity. A straightforward interpretation is that specialized food processing strategies involve correlated combinations of fine-tuned tooth shapes along the jaw, with function tied to the overall dental arrangement. As biomechanical studies often quantify single-tooth performances in piercing, slicing, or grinding (*Whitenack et al., 2011*; *Whitenack and Motta, 2010*; *Frazzetta, 1988*), our results suggest a need to complement those with whole-jaw testing paradigms (*Corn et al., 2016*; *Cooper et al., 2023*).

Intriguingly, we do not find any significant correlation between low-to-moderate genetic distances and heterodonty differences, implying absence of strong constraints that prevent closely related taxa from developing divergent heterodonty patterns. A similar observation was made for jaw shape differences, permitting stark divergences within relatively short evolutionary timespans (*López-Romero et al., 2023*). Conversely, disparity of tooth-level complexity, for most measures, appears to increase with genetic distance, suggesting significant phylogenetic constraints. This suggests that morphological adaptation tends to involve changes at the level of heterodonty rather than tooth morphology. Theoretical and experimental studies in mammals have demonstrated that both gradual and discrete heterodont tooth shape change can be achieved by a gradual modification of developmental parameters (*Harjunmaa et al., 2014*; *Salazar-Ciudad and Jernvall, 2010*). Thus, tinkering with odontogenesis in a global rather than local manner can generate adaptive phenotypic variation. We hypothesize that fine-tuning individual teeth without affecting their neighbors might be more difficult than altering jaw-level gradients of morphogens or developmental factors, which will impact downstream odontogenesis locally. Additionally, studies have shown that even functionally intertwined traits such as cichlid jaws can exhibit independent evolutionary dynamics (*Ronco and Salzburger, 2021*). In the context of shark dentition, this means that dignathic heterodonty in the absence of occlusal constraints should be evolvable within short timespans.

This would render dentition an accessible model system of hierarchical developmental modularity underlying a mosaic fashion of evolutionary change in a set of functionally or ontologically connected traits (*Zimm et al., 2023b*). Besides functional constraints and evolutionary contingency, differences in the frequency of variational patterns may reflect developmental biases (*Smith et al., 1985*). Given the weaker correlation of heterodonty differences with genetic distance compared to tooth-level traits, and a stronger association of heterodonties and ecological specializations, it is tempting to speculate that evolutionary change tends to developmentally originate from alterations in higher-level cues rather than from individually tinkering with low-level features. Evidence for this hypothesis comes from different lines of research: Developmental studies have revealed the explicit involvement of signal gradients from the jaw mesenchyme in establishing differences among mammalian teeth (*Tucker and Sharpe, 1999*; *Kavanagh et al., 2007*). Leveraging developmental transcriptomics, another recent study showed how evolution in one tooth will indirectly affect the developmental regulation of teeth in other positions (*Sémon et al., 2025*). At the micro- to meso-evolutionary level, morphometric correlations between different mammalian tooth types suggest regulation by shared, yet not identical, developmental factors (*Gómez-Robles et al., 2011*), although the degree of covariation varies across traits and species (*Laffont et al., 2009*).

Developmental and ontological nestedness might be a major biological principle (*Cantor et al., 2017*; *Zimm et al., 2023b*). It has been argued that the repetitive nesting of modules within generative networks can be considered a general strategy for generating complex yet diverse outcomes that transcends the domain of biology (*Solé and Valverde, 2020*). Theoretical research has emphasized that modularity will increase both robustness to undesirable variation and evolvability (*Klingenberg, 2004*; *Espinosa-Soto and Wagner, 2010*; *Wagner, 2008*), while different lines of evo-devo research have shown that functionally or developmentally integrated modules can evolve independently (*Ronco and Salzburger, 2021*; *Watanabe et al., 2019*; *Felice et al., 2018*; *Sémon et al., 2025*). In conclusion, teeth may be considered a model system to understand how nature adapts to environmental challenges not only by emergence and fine-tuning of hyperdiverse phenotypic modules (*Jernvall and Salazar-Ciudad, 2006*), but also by tinkering with their modular embedding in a morphological context across levels of organization.

## Materials and methods
### Shark dentition data acquisition and processing
We extracted entire lateral tooth outlines from selected shark species published on the j-elasmo database (j-elasmo: http://naka.na.coocan.jp/; outlines downloaded 06–2021). This database contains displays of entire erupted tooth rows of over 100 species. The selection for this study was based on the criteria of phylogenetic representativity (i.e. sampling from all extant major clades and avoidance of redundancy by sampling among phenotypically similar sister species) and completeness of dentition, aiming at high coverage of all types of dentitions among extant shark species. Dentitions of which too

many teeth overlapped visually were not used. However, negligibly overlapping teeth, that is teeth whose partial overlap with adjacent teeth was minor and did not obstruct important features such as cusps, were reconstructed by interpolation and comparison to neighboring teeth and included. More substantially visually obstructed or damaged teeth were excluded from the analysis. Occasionally visible minor damages such as small holes to the enameloid were manually corrected. The exception to the completeness criterion was *Pseudotriakis microdon*, which features extremely high counts of relatively small teeth. From this species, only a subset of teeth from different jaw positions was used. In addition, we excluded planktivorous sharks due to their highly specialized dentitions. As sharks keep generating teeth continuously, we defined tooth rows as the contiguous sequences of fully erupted teeth along the jaw, from meso/anterior to distal/posterior positions. Due to bilateral symmetry, only one jaw hemisphere was used. As the vast majority of shark teeth are blade-shaped and only feature negligible morphological information in bucco-lingual direction, we decided that 2D lateral views suffice for the purposes of our study. Tooth shape extraction was performed using custom-made tools that automatically identify tooth boundaries and manual segmentation where necessary. Tooth size was not taken into consideration within the scope of this study, because it cannot be explicitly included in the set of shape difference descriptors we used. In other words, size differences can be considered another, independent, dimension of phenotypic features that may differ between teeth. Since the shape of the basal part of the tooth crown tends to show less morphological-functional fine-tuning than the upper part that is exposed to nutrition and at times shows a less defined, undulating, or porous transition to the jaw mesenchyme, we decided to only consider the upper dental outline. In preliminary analyses for which the entire tooth outlines were considered, we had found that morphometric patterns were in part driven by the basal rather than the upper part of the tooth crowns. The segmentation point between upper and lower parts of the outlines was defined by (1) a visual transition of material, otherwise by (2) the lowest concave or most concave lateral point if a visible inflection could be discerned or (3) the most distant pair of outline points in the lower part. Outline point numbers (1000 per tooth) were equalized by interpolation or data reduction in order to ensure comparability across teeth.

## Ecological information

In order to be able to associate tooth phenotypic information with potential ecological function, we collected proxy features that were widely available in databases and publications. The trophic level was estimated based on published information from stomach contents or pre-calculated trophic scores as referenced in FISHBASE (*Froese and Pauly, 2010*), shark references (*Pollerspöck and Straube, 2014*), and a number of individual sources (see Sharks_eco_refs.xlsx for references *Cortes, 1999*; *Bizzarro et al., 2017*; *Ba et al., 2013*; *Ba et al., 2013*; *Stevens and Cuthbert, 1983*; *Yano and Musick, 1992*; *Osgood and Baum, 2015*; *Barnett et al., 2013*; *Horie and Tanaka, 2000*; *Park et al., 2019*; *Kamura and Hashimoto, 2004*; *Fergusson et al., 2007*; *Kindong et al., 2021*; *Li et al., 2014*; *Vaudo and Heithaus, 2011*; *Huveneers et al., 2007*; *Burke et al., 2024*; *Dunn et al., 2013*; *Carlisle et al., 2021*; *Yano et al., 2003*; *Baremore et al., 2008*; *Baremore et al., 2010*; *Campagno, 1990*). In the few cases where suitable trophic information was not available, the phylogenetically closest species for which sufficient records were accessible were supplied instead. In the occasional case of conflicting values, the more detailed, higher-quality, or more clearly documented of the available sources was used preferentially. In addition, we assigned recorded prey items to larger trophic guilds. Piscivory was not assigned as it is highly unspecific with respect to prey size and trophic level, and because nearly all species include fish into their diet. Another specific diet, planctivory, was omitted, as the three plankton-feeding sharks feature very specialized dentitions, which tend to occupy separate parts within morphospaces (*Bazzi et al., 2018*). Taken together, we used the following categories: TROPH: average trophic level, VERT: non-osteichthyan vertebrate prey, CEPH: cephalopod prey, MOLL: other molluscan prey, CRUST: crustacean prey, VERM: further small usually worm-like invertebrate prey, OMNI: degree of omnivory or number of documented food categories. Body length, depth, and habitat information was compiled using FISHBASE (*Froese and Pauly, 2010*), Shark references (*Pollerspöck and Straube, 2014*), and Sharks of the World (*Campagno et al., 2005*). Extreme body length values were neglected as exceptions, or possibly overstated reports, and the documented ranges for mature male and female individuals were used. This means we excluded unconfirmed reports and measurements from single specimens that substantially exceeded the bulk

of habitual measurements, suggesting unreliable or exaggerated claims, or rare outliers unrepresentative of the typical ecological niche of the respective taxon. Unless noted differently, length values used in our analyses were calculated as the average between upper and lower range limits for females and males, respectively, and the average of those. We used the following categories: SIZE: body length, as calculated above, $S_{MIN}$, $S_{MAX}$: reported extremes of adult/fertile individuals, $S_{HATCH}$: size at hatching or birth. Depth was annotated similarly; we averaged between the upper and lower range limits that were reported, not considering exceptional reports: DEPTH: general depth of occurrence, $D_{LOW}$: lower depth limit, $D_{HIGH}$: upper depth limit, $D_{RANGE}$: difference between $D_{LOW}$ and $D_{HIGH}$ We also noted whether shark species were associated with specific habitat types, as within the set of references used. For this assignment, we searched for habitat descriptor terms in encyclopedic literature (as given above) that were not explicitly mentioned as exceptional occurrence. This way, we avoided the need to define arbitrary limits between complementary ecological descriptors. In this paper, we used the following habitat categories: SHORE: occurrence near shore line, SHELF: occurrence along the continental shelf zone, REEF: reef habitat, OCEAN: open sea, DEEP: deep sea, BENT: benthic habitat, NECT: nektonic habitat. Albeit potentially biased in multiple ways, this compilation of data represents what is currently known and available in the published literature.

## Phylogenetic analysis

To build the phylogenetic tree, the following, slowly evolving and commonly used genetic markers were selected: COI, cytB, NADH2, the ribosomal 12 S, 16 S genes (with full sequences of 12S+tRNA-Val+16 S where available), and rag-1. The choice was made based on availability and a previously published phylogeny (*Vélez-Zuazo and Agnarsson, 2011*). See the NCBI accession numbers in the *Supplementary file 1*. The sequences were concatenated in the specified order and aligned with MUSCLE (*Edgar, 2004*) (through the program Unipro UGENE *Okonechnikov et al., 2012*) with default parameter options. Alignment regions with gap values higher than 95% were trimmed. The phylogenetic tree was finally built using the program PHYLIP (*Baum, 1989*; *Revell and Chamberlain, 2014*), generating a neighbor-joining tree. The neighbor-joining method was used herein because it aligns with standard approaches in comparable studies that involve similar genetic datasets, allowing for direct comparability to previously published phylogenetic analyses, and because it provides fast computation even for large sets of data and is appropriate for clustering the species relationships when the genetic data is incomplete or heterogeneous. Kimura's two-parameter model (K2P) was used to compute a distance matrix. We used a bootstrap of 1000, Seed values of 5 and the Majority Rule (extended) as a consensus-type choice were applied. The final tree was visualized using TreeViewer (*Bianchini and Sánchez-Baracaldo, 2024*). For two species, corresponding genetic information was not available: *Oxynotus centrina* and *Heterodontus japonicus*. In order to build the phylogenetic tree, *O. centrina* was added, integrating the phylogenetic analysis of *Straube et al., 2015*; *Straube et al., 2015*, that is including the same relative branch lengths as provided therein, while *H. japonicus* was assumed to have a position very close to *H. zebra*. Although both species belong to the same genus, we decided to include two specimens of Heterodontidae to provide more than one data point for this clade. Where used, taxonomic categories were assigned according to literature. However, commonly established taxonomic families that resulted in paraphyly were not assigned, and polyphyletic units were assigned independently. Intermediate taxonomic levels (super-families to infra-orders) were determined based on the phylogenetic tree at hand. Tree branch lengths normalized by maximal and minimal distance between species pairs were added up to quantify genetic distances (dG).

## Tooth comparison

In the absence of a methodological gold standard to quantify phenotypic similarity between tooth pairs, we devised a set of six measures, thus capturing different aspects of shape, *Figure 1—figure supplement 1*. These similarity measures are then deployed to quantify heterodonty by calculating average pair-wise distances between teeth according to the respective heterodonty definitions.

Partial Procrustes Alignment: To minimize the part of shape difference attributed to relative placement, we performed an incremental shape rotation by up to $\pm \pi/8$, a shift along x and y axes by up to 10 %, and a size change by up to $\pm 25$ %. These ranges were determined as sufficient in precursory tests with a subset of shape comparisons. The initial size difference correction was done in two ways, by normalization by total outline length and normalization by total tooth area, and the lower resulting

distance was kept. Tooth area was defined by the outline and a straight line connecting its start and end points. Both tooth outlines were centered on their centroids. For the first three shape distance measures, the Procrustes-aligned configuration issuing the smallest distance between the outline pairs was then considered their definitive distance.

(a) Euclidean mean distance (EMD): For each pair of morphologies, we calculated the mean distance between every point (x,y) along one outline L1 and the physically closest outline point (X,Y) of the other morphology L2, irrespective of its relative position. This procedure was conducted both ways.

$$EMD_{L1,L2} = \frac{\sum_{i=1}^{n_{L1}} min_{j=1}^{n_{L2}} \sqrt{(x_i - X_j)^2 + (y_i - Y_j)^2}}{n_{L1}} + \frac{\sum_{j=1}^{n_{L2}} min_{i=1}^{n_{L1}} \sqrt{(X_j - x_i)^2 + (Y_j - y_i)^2}}{n_{L2}} \tag{1}$$

(b) Homologous Euclidean outline distance (HED): while the EMD does not make any assumptions about homology, this method compares identical relative positions along two outlines, thus representing pseudo-homology, with distant similarities to semi-landmark methods. $i = \{1,..,n_{L1}\}$ and $j = \{1,..,n_{L2}\}$ are the respective outline points in the two teeth.

$$HED_{L1,L2} = \frac{\sum_{i=1}^{n_{L1}} \sqrt{(x_i - X_k)^2 + (y_i - Y_k)^2}}{n_{L1}} + \frac{\sum_{j=1}^{n_{L2}} \sqrt{(X_j - x_l)^2 + (Y_j - y_l)^2}}{n_{L2}};$$
$$k = j_{min|i/ni - j/nj|}; l = i_{min|i/ni - j/nj|} \tag{2}$$

For our specific purposes, $n_{L1}=n_{L2}=n$, leading to a simplified formula:

$$HED_{L1,L2} = 2\frac{\sum_{i=1}^{n} \sqrt{(x_i - X_i)^2 + (y_i - Y_i)^2}}{n} \tag{3}$$

(c) Superimposed area overlap (SAO): This method calculates the ratio between the counts of overlapping and non-overlapping parts of the overlaid tooth shapes. The lower boundary delineating the area is defined by a straight line connecting the start and end points of the outlines. To calculate area overlap, both shapes were rasterized into a number of small squares S (i.e. pixels). Before applying Partial Procrustes Alignment, the maximum x and y distances were used to discretize both axes into 100 units, respectively.

$$SAO_{L1,L2} = 2\frac{\sum (S_{L1 \cap L2})}{\sum (S_{L1}) + \sum (S_{L2})} \tag{4}$$

(d) Discrete Cosine Fourier distance (DFD): Similar shapes are expected to be defined by sets of similar Fourier coefficients. For this measure, we applied discrete cosine Fourier transformation on the tooth outlines, which incrementally approximates semi-outlines by superimposing cosine lines. The distance is then calculated as the Euclidean distance between all coefficients (for the first 24 harmonics) Morphological distances between two shapes $i$ and $j$ were quantified by calculating Euclidean distances between the values $z(\epsilon)$ of the Fourier coefficients $\epsilon$, $n_\epsilon$ being the number of coefficients at 24 harmonics:

$$DFD_{L1,L2} = \sqrt{\sum_{\varepsilon=1}^{n\varepsilon} (z_{L1}(\varepsilon) - z_{L2}(\varepsilon))^2} \tag{5}$$

(e) Outline angle sum distance (OAD): Outline angles can be calculated between triplets of subsequential outline points. We used the sum function of surface angles for n=100 equidistant outline points (i.e. after point number reduction in order to reduce noise) as a descriptor of shape. We then overlaid the outline angle sum functions $af(i)$ of two tooth outlines L1 and L2, starting from the same value, and calculated the area between them. Pairs of similar teeth are expected to show similar functions $af(i)$ and low differences in between.

$$OAD_{L1,L2} = \int_0^n |af(i)_{L2} - af(i)_{L1}| \, di \tag{6}$$

(f) Angle Function Discrete Cosine Fourier distance (ADD): In analogy to the DFD, we approximate the specific angle sum function of the OAD by means of discrete cosine Fourier transformation. We then use the Euclidean distance between the resulting Fourier coefficients to describe differences between a given pair of outline functions.

$$ADD_{L1,L2} = \sqrt{\sum_{\varepsilon=1}^{n\varepsilon} \left(af_{L1}(\varepsilon) - af_{L2}(\varepsilon)\right)^2} \tag{7}$$

## Heterodonty measures

Heterodonty per dentition was then calculated as the average of all distances, as defined above, between any pair of teeth in consideration. We distinguished between three different heterodonty measures: (a) sequential monognathic heterodonty (HMS): shape difference between neighboring teeth, (b) total monognathic heterodonty (HMT): shape difference between any pair of teeth within the same jaw, (c) dignathic heterodonty (HDG): shape difference between pairs of teeth at approximately opposite positions on upper and lower jaws. Where the number of teeth differed between the opposing jaws, relative positions were used, eventually causing the same tooth to be compared to more than one tooth in the opposing jaw if the latter harbored a larger number of teeth. As introduced above, a schematic of these measures can be seen in *Figure 1—figure supplement 1*. Note that in sharks, no dental occlusion occurs, allowing a higher degree of morphological freedom than in many mammals. In addition, we also recorded the maximum distance between any two teeth within a given jaw (HMX). As a jaw-level descriptor, measures were normalized by division by the respective number of tooth comparisons.

In the following measures, i and j refer to teeth within a respective tooth row or opposing tooth rows for the case of dignathic heterodonty; x denotes any tooth shape descriptor.

$$HMS = \frac{\sum_{i=1}^{n} |x_i - x_{i-1}| + |x_i - x_{i+1}|}{2n} \tag{8}$$

$$HMT = \frac{\sum_{i=1}^{n} \sum_{j=1}^{n} |x_i - x_j|}{n^2} \tag{9}$$

$$HDG = \frac{\sum_{i=1}^{ni} |x_i - x_k|}{ni} + \frac{\sum_{j=1}^{nj} |x_j - x_l|}{nj} ; \quad jaw_{i,l} \neq jaw_{j,k}; \quad k = j_{min|i/ni - j/nj|}; \quad l = i_{min|j/nj - i/ni|} \tag{10}$$

$$HMX = max |x_i - x_j| ; \quad jaw_i = jaw_j \tag{11}$$

Unless declared otherwise, we calculated the values of heterodonty as the average of all six distance measures devised, in order to minimize potential biases introduced by the choice of a specific method. For interspecies comparison, all values were normalized by the global maxima and minima, respectively.

## Tooth complexity measurements

As for shape distances between pairs of teeth, there is no commonly accepted gold standard method to quantify complexity, even more as it may refer to different features. This is why we devised a range of different methods, as schematically shown in *Figure 1—figure supplement 2*. For several analyses, we pooled similar complexity methods, such as outline-based or angle-based methods. Where not stated otherwise, we calculated total complexity as the sum of all introduced measures. In the displayed formulae, i and j denote outline points of the same tooth, unless explicitly stated otherwise.

(a) Coarse-grained cuspidity (CUSP1): the number of larger cusps.

(b) Fine-grained cuspidity (CUSP2): the number of minor cusplets. The difference to the previous measure was defined, unavoidably, by an arbitrarily chosen relative size threshold: while the largest cusp was always considered major, cusps were considered minor if their higher col was below 2% of the total length or if they clearly constituted a serration pattern on larger cusps.

(c) Outline-to-area ratio (OAR): The total length of the outline was divided by the total area.

(d) Outline-to-centroid size ratio (OCR): instead of area, outline length was divided by the centroid size.

$$OCR = \frac{\sum_{i=2}^{n} \sqrt{(x_i - x_{i-1})^2 + (y_i - y_{i-1})^2}}{\sqrt{\sum_{i=1}^{n} \sqrt{(x_i - \bar{x})^2 + (y_i - \bar{y})^2}}} \tag{12}$$

(e) Outer/inner circle ratio (OIR): the area of the largest circle inscribed in the outline was divided by the area of the smallest escribed circle encompassing the tooth outline. To prevent a few large ratios from skewing the distribution, we defined a cutoff of 25. This measure captures differences in eccentricity.

(f) Discrete Cosine Fourier coefficients sum (DFS): due to the definition of Fourier analyses, the size of its coefficients correlates with the eccentricity, feature diversity, and difference to a simple round shape. As such, the total sum of Fourier coefficients z for a given outline can be used as a proxy of information required to describe shapes, i.e. shape complexity.

$$DFS = \sum_{\varepsilon=1}^{n\varepsilon} \sum_{i=1}^{4} |z_{i\varepsilon}| \quad ; \quad i : coefficients \quad ; \quad \varepsilon : harmonics \tag{13}$$

(g) Angle sum (ANS): sum of all surface angles (here defined by three points along the outline; angles different from 180° / π rad yield higher values.) for NR different resolutions R (defined by the total numbers of equally spaced outline points $n_R$). For our angle-based measures, we used six different resolutions with $n_R = n/(5(2^{R-1}))$ for R={1,..,6}.

$$ANS = \sum_{R=1}^{NR} A(R) * (NR)^{-1} ; \quad A(R) = \sum_{i=2}^{n_R-1} |\angle(x_R(i-1)|x_R(i)|x_R(i+1)) - \pi| \tag{14}$$

(h) Angle sum cadence (ASC): a measure of difference between the angle counts across different resolutions. This reflects the fact that repetitive traits will feature large differences between resolutions, while outlines with differently sized traits will show less difference. In general, the latter case will be considered less complex, as it contains less information.

$$ASC = \sum_{R=2}^{NR} (A(R) - A(R-1))(NR-1)^{-1} \tag{15}$$

(i) Angle disparity (AND): In a similar vein, we measure, for different resolutions, the diversity of angles between pairs of adjacent points on the outlines. Larger diversity is associated with morphological complexity.

$$AND = \sum_{R=1}^{NR} \sum_{i=2}^{n_R-1} \sum_{j=2}^{n_R-1} (\alpha(R,i) - \alpha(R,j))(NR)^{-1}(n_R-2)^{-2} ; \quad \alpha(R,i) = \angle(x_R(i-1)|x_R(i)|x_R(i+1)) \tag{16}$$

(j) Orientation patch count (OPC) **Evans et al., 2007**: we count the number of contiguous outline streaks that are delimited by a change in absolute direction. Absolute direction is defined by the vectors between subsequent outline points and discretized to absolute partitions of a circle, that is a change of direction would correspond to a change of partition and an increase of the count. For this measure, we used different partitions (2,4,8), different rotations of the coordinate system (no rotation, rotation by half a partition and by quarter partitions for the lowest partition number), and the different outline resolutions listed above.

## Phenotypic distance between species

In addition, we calculated total phenotypic distances DP between species pairs (i,j). This is to serve as a test to see how overall similarity would scale with genetic distance, as calculated above. For this measure, teeth of comparable relative jaw positions in two species were compared in a manner analogous to the dignathic heterodonty measure. ni and nj are the total number of teeth per row for the two species, respectively.

$$DP(i,j) = \frac{\sum_{i=1}^{ni} |x_i - x_k|}{ni} + \frac{\sum_{j=1}^{nj} |x_j - x_l|}{nj} ; \quad jaw_{i,l} = jaw_{j,k}; \quad k = j_{min|i/ni - j/nj|}; \quad l = i_{min|j/nj - i/ni|} \tag{17}$$

## Statistical analyses

Tooth mean shapes were calculated by averaging the discrete cosine Fourier coefficients within the set of chosen specimens and inversely reconstructing the tooth shape. These operations were performed using the dfourier function contained within the R package Momocs *Bonhomme et al., 2014*. This package was also used to perform shape-based PCA.

Canonical correlation analysis based on varying sets of traits was conducted using the R package Cancor.

In order to quantify the phylogenetic signal, we took advantage of several of the most frequently used methods: Abouheif's c-mean, Moran's P, Pagel's Lambda, Blomberg's K. We used available R packages to conduct the analyses: abouheif.moran, moran.idx from the adephylo library and phylosig from the phytools library.

We used common R functions (cor, t.test, wilcox.test) as well as the linear regression function via STATS of gnuplot in order to calculate correlation coefficients and p-values. Wilcoxon tests are Mann-Whitney U tests. All statistical tests were two-sided unless stated differently.

## Data visualization

We used gnuplot (version 5.2) and R (v.4.0.3; basic plot functions and functions from the ggplot2 library) to plot data.

## Acknowledgements

The authors are grateful to Fumio Nakagawa for his role in building the j-elasmo database as well as for helpful interactions, and to Arthur Gairin-Calvo, Fidji Berio, Miguel Brun-Usan, and Samuel Ginot, as well as two anonymous reviewers, for critical and constructive comments. This work was supported by a French ANR grant (ANR-21-CE02-0015 PLASTICiTEETH to NG) and a Deutsche Forschungs-gesellschaft DFG research fellowship (ZI1809/1-1:1, Proj.432922638 to RZ). A previous version of this manuscript is available as a preprint: *Zimm et al., 2024*; https://doi.org/10.1101/.

## Additional information

### Funding

| Funder | Grant reference number | Author |
| --- | --- | --- |
| Deutsche Forschungsgemeinschaft | ZI1809/1-1:1 Proj.432922638 | Roland Zimm |
| Agence Nationale de la Recherche | ANR-21-CE02-0015 PLASTICiTEETH | Nicolas Goudemand |

The funders had no role in study design, data collection and interpretation, or the decision to submit the work for publication.

### Author contributions

Roland Zimm, Conceptualization, Data curation, Software, Formal analysis, Funding acquisition, Validation, Investigation, Visualization, Methodology, Writing – original draft, Writing – review and editing; Vitória Tobias Santos, Formal analysis, Investigation, Visualization, Methodology, Writing – original draft, Writing – review and editing; Nicolas Goudemand, Conceptualization, Supervision, Funding acquisition, Writing – original draft, Project administration, Writing – review and editing

### Author ORCIDs

Roland Zimm https://orcid.org/0000-0002-0388-7052
Vitória Tobias Santos https://orcid.org/0000-0001-5958-9700
Nicolas Goudemand https://orcid.org/0000-0002-2956-5852

### Decision letter and Author response

Decision letter https://doi.org/10.7554/eLife.107406.sa1
Author response https://doi.org/10.7554/eLife.107406.sa2

## Additional files

### Supplementary files

Supplementary file 1. List_NCBI_refs.xlsx. This file lists all NCBI sequence references used to reconstruct the shark phylogeny.

Supplementary file 2. Sharks_eco_refs.xlsx. Overview of ecological information per species and their respective references.

MDAR checklist

### Data availability

Shark morphological data was taken from a publicly accessible shark tooth picture collection, with approval of the author. Publicly available NCBI sequence data were used in the analysis and are listed in the SI. All data pertaining to the study as well as supporting files and relevant code has been uploaded to an open repository: https://github.com/RolandZimm/shark_heterodonty (copy archived at *Zimm, 2025*) and https://doi.org/10.5281/zenodo.14545383.

The following dataset was generated:

| Author(s) | Year | Dataset title | Dataset URL | Database and Identifier |
|---|---|---|---|---|
| Zimm R | 2024 | RolandZimm/ shark_heterodonty: Sharkheterodonty_ release231224 | https://doi.org/ 10.5281/zenodo. 14545383 | Zenodo, 10.5281/ zenodo.14545383 |

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
