## [Editor Report]

This important study combines morphological and genetic information of teeth and dentitions of extant sharks with a novel morphometric tool combination and phylogenetic information to better understand the relevance of hierarchical organization of functional traits to identify trait adaptability. Indicating developmental and ontological nestedness, the overall results are convincing and have the potential to be fundamental to the application of dental traits for paleoenvironmental and palaeoecological reconstruction in sharks, vertebrates in general.

---

## [Decision Letter]

**Decision letter after peer review:**

Thank you for submitting your article "Integration of multi-level dental diversity links macro-evolutionary patterns to ecological strategies across sharks" for consideration by *eLife*. Your article has been reviewed by 2 peer reviewers, and the evaluation has been overseen by a Reviewing Editor and Detlef Weigel as the Senior Editor.

Essential Revisions:

1) Please take into account the reviewer comments to improve the accessibility of the manuscript.

2) Please expand the discussion regarding the implications of your results.

*Reviewer #1 (Recommendations for the authors):*

Living sharks are characterized by exceptional dental diversities on all levels of hierarchy from individual teeth to complete dentitions. Such proxies are not only used for taxonomic identification but also for ecological function and feeding adaptations. What also is evident is that teeth within jaws might differ morphologically from mesial to distal or even between upper and lower jaws indicating varying degrees of monognathic and dignathic heterodonties, respectively. It already has been demonstrated that jaw geometry, cranial shape, and musculature among other macro-features are linked to feeding strategies. This highlights that different traits result in adaptive interplays, certainly also underlying the evolutionary success of sharks.

In this study, the authors analysed morphological features ranging from isolated teeth to whole dentitions within a large ensemble of species encompassing all extant shark orders applying a novel combination of morphometric approaches and ecological and life-history traits to better understand the specific relevance of hierarchical organization of functional traits. In doing so, the authors intend to track these heterodonties on all levels as a function of genetic distance within a strict phylogenetic framework to quantify trait adaptability. Understanding hierarchical organizations of functional traits is not only of importance to biologists, but also to palaeontologist, thus representing a cross-disciplinary study.

The analyses are based on 2D shape data from 51 extant shark species representing all orders derived from J-elasmo (http://naka.na.coocan.jp/). For reaching the goals of this study, the authors established differences between teeth, neighbouring teeth within the same jaw, i.e., sequential-monognathic heterodonty, between all teeth within the same jaw, i.e., total-monognathic heterodonty and between teeth from same relative positions in opposing jaws, i.e., dignathic heterodonty, identified the ecology of the various species, conducted phylogenetic analyses using different genetic markers, used novel tooth and dentition measures and employed robust statistic methodologies to calculate mean tooth shapes. However, it would be easier to understand the overarching goal and the individual aims if they would have been clearly formulated at the end of the introduction simultaneously formulating corresponding hypotheses.

The undisputed significance of this study is in the combination of the varying morphological levels (individual teeth to complete dentitions) and the methodological approaches, which enable the authors to gain quantitatively well-supported insights into the importance of dental heterodonties in sharks.

The results of this study demonstrate that dental variations within and between jaws obviously is not independent, with sequential heterodonty being significantly lower and dignathic heterodonty being significantly higher in squalomorph sharks. In highly specialized dentitions as those of extant hexanchiforms (Squalomorphii), the highest levels of both monognathic and dignathic heterodonty occur, while in various squalomorphs (e.g., Squalus, Squatina) and galeomorphs (e.g., Mustelus, Nebrius) these heterodonties are lowest.

The high heterodonty found in extant hexanchids is considered to be functional and conserved throughout the proximate fossil record (lines 185-186). However, what does "proximate fossil record" mean. In stem hexanchiforms such as Notidanoides from the Jurassic, seemingly only sequential heterodonty is developed with dignathic heterodonty being absent compare Kriwet and Klug 2014 [Dental patterns of the stem-group hexanchoid shark, Notidanoides muensteri (Elasmobranchii, Hexanchiformes)]. Thus, there seems to be a constraint in the evolution of hexanchiforms towards dignathic heterodonty. Including fossil taxa could increase our understanding of evolutionary pathways of such adaptive traits better than just using extant forms to identify possible constraints.

Conversely to squalomorphs, galeomorph sharks have a more gradual (sequential) heterodonty. Mapping all information derived from the shape analyses across all hierarchical levels indicates that the various heterodonty patterns and tooth-level complexities evolved independently several times based on information derived from extant taxa. Importantly, this approach reveals that no significant positive correlation between genetic distance and heterodonty differences for genetically close or moderately distant species exist, but genetically most distant species display significantly higher heterodonty differences. This indicates that heterodonty can change relatively unconstrained and thus is highly evolvable (lines 117-118). There seems to be no strong constraints preventing closely related taxa from developing divergent heterodonty patterns (line 273-275).

However, this seems odd. If only genetically distant species exhibit significantly higher heterodonties, why can heterodonty then change relatively unconstrained in low-to-moderately genetically related taxa? I would expect such a process if the pattern also is significant in closely related species. The study is based on 51 extant species, each obviously represented by a single specimen. While this rather large number of species and corresponding dental information, it does not catch the intraspecific and possible interspecific dental variations within extant taxa, let alone in fossil taxa. This information, however, could provide additional information about evolutionary pathways and constraints!

In lines 121-133, the authors state some tooth-level complexity measures might serve as a moderately better proxy for relatedness than heterodonty. Which measures are these? It would be useful to list these.

Interestingly, including ecological features, especially depth distribution in the analyses results in a better separation of the two superorders indicating that heterodonty and tooth complexity patterns result from adaptive morphological changes. Thus, the significantly higher dignathic heterodonty found in many squalomorphs would be the result of deep-water adaptation. But it remains ambiguous why this is. And moreover, dignathic heterodonty is absent in deep-water galeomorphs, such as, e.g., Apristurus. I would suggest to list such exceptions and to discuss possible reasons for presence / absence of dignathic heterodonties in relation to habitat in more detail.

In lines 174-175, the authors state that dental variation evolves dynamically within sharks. This result seemingly relies on the combination of the shape analyses with phylogenetic information. It remains unclear what dynamically means. Moreover, such a statement should be supported be specific trait analyses.

The authors continue to compare individual tooth shape development in sharks with that in mammals, stating that Hox genes are important in differentiating tooth classes in mammals, while the role of Hox genes in sharks remains ambiguous. A major difference between both groups, however, is the presence of a dental lamina in sharks, which is important in tooth development, and which is absent in mammals. Could it that this difference is more important than Hox genes? Compare, e.g., various publications by Gareth Fraser (see https://www.fraser-lab.net/publications.html).

According to the results, habitats correlate with feeding habits representing a proxy for a species' primary food sources. Based on this, two discrete clusters representing major ecological-functional strategies are identified, i.e., shallow-water habitats, crustacean/small invertebrate diets and open-sea habitats, large preys, with a sub-cluster of deep-sea cephalopod feeders. However, I wonder if this is not an oversimplification? Nevertheless, the results show that monognathic heterodonty is stronger correlated with ecological traits than dignathic heterodonty.

The study importantly shows that evolutionary change tends to developmentally originate from alterations in higher-level cues rather than from individually tinkering with individual teeth. This indicates that developmental and ontological nestedness could be a major biological principle.

Overall, the authors achieved the targeted goals based on the data set used here. But as stated above, inclusion of more specimens of each species and more species per clade as well as extinct taxa might provide additional information about inter- and intraspecific variations in heterodonty patterns through time and corresponding constraints. This also might better show, if heterodonty patterns are really highest in genetically distant taxa.

Nevertheless, this study is the first one using combined methodological approaches to establish the significance of heterodonty patterns on various hierarchical scales and thus certainly will serve a roadmap for future studies into the development of teeth and dentitions in sharks, and also beyond.

This is a very important study. But I have to admit that it was quite difficult to read and to follow your arguments. Therefore, I would like to make some additional comments.

- Include in the abstract ALL findings (results) of your study.

- State clearly all the goals. It was quite difficult to really identify all goals of the study as you only mention "Here, we elucidate such macro-patterns in the light of environmental and life-history traits, applying a novel morphometric tool combination. This may critically contribute to understanding the specific relevance of hierarchical organization of functional traits across an entire vertebrate class, a significant question across ecology, palaeontology and evolution". Only by reading all the manuscript and focusing on your results it was possible to really understand what the different goals were. I also would expect some research hypothesis that you could directly address in the results and Discussion sections.

- Please use systematic names instead of G1 and G2 in the text! I had to refer to the various figures to find out the colour code for the two major groups in one figure, then refer to the next figure to correlate the colour code / systematic name with G1 and G2, respectively to understand what you meant.

- I would strongly suggest explaining the results of the graphs shown in Figure 4C in more detail in the text. In many instances, you simply make a statement without properly discussing it.

- What makes the manuscript difficult to read is the structure and the sometimes jumping around between topics. I would strongly suggest to alter the structure and address the different research questions / hypotheses so that there is a direct link between the goals indicated and the results and discussions.

*Reviewer #2 (Recommendations for the authors):*

Zimm et al. investigated the relationships between dental traits, including heterodonty and complexity, with various aspects of shark biology, analysing a range of data from pre-existing sources in a novel and creative way. They find that traits coded on an intermediate level e.g. within a jaw show a low phylogenetic signal and a relatively strong relationship with ecology, suggesting significant adaptability of these traits. Particularly impressive is the quantification of these dental traits. The number of methodological approaches used to measure complexity and heterodonty is remarkable, and well-reasoned. There is excellent attention paid to the context of the study, given work that has been done previously in mammals, and the phylogenetic context is given through the genetic distance approach, which is important, and not always done in similar trait analyses for other clades.

The most significant aspect of the paper that could be expanded is the discussion of the implications of these results. In the abstract, introduction and discussion, the fossil record of sharks is mentioned, and one of the papers frequently cited in the introduction (Eronen et al., 2010) describes the use of traits, including tooth shape to reconstruct environments through the application of ecometrics. This has been done in mammals, but not, as far as I'm aware, using such a range of scales as in this study. Given the attention given to potential palaeoecological applications of the results here, I would suggest there could be further discussion of this. In particular, I wonder if a discussion of the availability of tooth rows to enough of an extent for measurement of heterodonty would be useful, as well as a consideration of the potential use of community-level shark traits for an assessment of water depth, for example.

In addition to the comments in the public review, I noticed a few small details that I would suggest addressing, although in general I thought the presentation of the work was excellent.

• In general, it would be helpful to restate the definitions for acronyms in all figure captions. There are a lot, understandably, but for a reader it would be helpful to have the definitions repeated in the captions e.g. for HMS, HMT, HMX, HDG.

• Figure 1 – Species names should be italicised in the figure. Squalomorphii and Galeomorphii are italicised in the caption but not in the rest of the text.

• Introduction – final paragraph could be expanded and made more specific about the methods and rationale

• Page 3, line 98 – I suggest making the clause "the main outliers…" into its own separate sentence, and I wonder if it would be better to use the full name rather than "squalean" which I don't think was previously defined

• Figure 2 – This may be personal preference, but I think it would be good to edit the figure so that the tip names are italicised and separated by a space rather than an underscore e.g. Carcharhinus sealei. The alignment of the bottom part of the figure between the different components is different and would be worth editing in the software used to make the figure. Should Selachimorpha be italicised in the caption? Other clade names in the text and in the figure are not.

• Section 4 (corresponding to Figure 5) – the choice of colour here is quite misleading, given that the same colours (or very similar) have been used to differentiate between the Squalomorphii and Galeomorphii in previous figures. Given that these groups do not simply correspond to the two major clades of sharks considered in Figure 4 for example, I think it would be preferable to use different colours to denote G1 and G2 in Figure 5. It would also be good to have included in the supplementary material the group assignment for each shark species included in the study. I note that this is included in the Github, but not in the spreadsheets available in the Supplementary Material.

• Line 365 – how many taxa was this substituting of close relatives used? It would be good to have an estimate for this.

• Line 374 – how is "extreme" defined? Similarly, the "exceptional reports" of line 379

• The list of ecological descriptors would be more useful if it were distributed in the text according to where the different categories are mentioned. For example, the prey categories could be listed at Line 368

• In the methods, you mention that the complexity is defined as the sum of the constituent measures. I'm not sure if I'm misunderstanding, but are the different measures actually comparable in this way? I.e. are the different measures on the same scale in a way that would allow them to be summed? I'm struggling to tell from the raw data available in the Github, but if not, this would be quite problematic.

---

## [Author Response]

Essential Revisions:Reviewer #1 (Recommendations for the authors):Living sharks are characterized by exceptional dental diversities on all levels of hierarchy from individual teeth to complete dentitions. Such proxies are not only used for taxonomic identification but also for ecological function and feeding adaptations. What also is evident is that teeth within jaws might differ morphologically from mesial to distal or even between upper and lower jaws indicating varying degrees of monognathic and dignathic heterodonties, respectively. It already has been demonstrated that jaw geometry, cranial shape, and musculature among other macro-features are linked to feeding strategies. This highlights that different traits result in adaptive interplays, certainly also underlying the evolutionary success of sharks.In this study, the authors analysed morphological features ranging from isolated teeth to whole dentitions within a large ensemble of species encompassing all extant shark orders applying a novel combination of morphometric approaches and ecological and life-history traits to better understand the specific relevance of hierarchical organization of functional traits. In doing so, the authors intend to track these heterodonties on all levels as a function of genetic distance within a strict phylogenetic framework to quantify trait adaptability. Understanding hierarchical organizations of functional traits is not only of importance to biologists, but also to palaeontologist, thus representing a cross-disciplinary study.The analyses are based on 2D shape data from 51 extant shark species representing all orders derived from J-elasmo (http://naka.na.coocan.jp/). For reaching the goals of this study, the authors established differences between teeth, neighbouring teeth within the same jaw, i.e., sequential-monognathic heterodonty, between all teeth within the same jaw, i.e., total-monognathic heterodonty and between teeth from same relative positions in opposing jaws, i.e., dignathic heterodonty, identified the ecology of the various species, conducted phylogenetic analyses using different genetic markers, used novel tooth and dentition measures and employed robust statistic methodologies to calculate mean tooth shapes. However, it would be easier to understand the overarching goal and the individual aims if they would have been clearly formulated at the end of the introduction simultaneously formulating corresponding hypotheses.The undisputed significance of this study is in the combination of the varying morphological levels (individual teeth to complete dentitions) and the methodological approaches, which enable the authors to gain quantitatively well-supported insights into the importance of dental heterodonties in sharks.The results of this study demonstrate that dental variations within and between jaws obviously is not independent, with sequential heterodonty being significantly lower and dignathic heterodonty being significantly higher in squalomorph sharks. In highly specialized dentitions as those of extant hexanchiforms (Squalomorphii), the highest levels of both monognathic and dignathic heterodonty occur, while in various squalomorphs (e.g., Squalus, Squatina) and galeomorphs (e.g., Mustelus, Nebrius) these heterodonties are lowest.The high heterodonty found in extant hexanchids is considered to be functional and conserved throughout the proximate fossil record (lines 185-186). However, what does "proximate fossil record" mean.

With “proximate”, we were trying to encapsulate the fact that the hitherto discussed paleontological horizon was limited to the Eocene which, compared to the phylogenetic divergence ages of the main shark lineages, is a relatively recent epoch. However, we agree with the reviewer that this attribute adds confusion and have consequently removed it.

In stem hexanchiforms such as Notidanoides from the Jurassic, seemingly only sequential heterodonty is developed with dignathic heterodonty being absent compare Kriwet and Klug 2014 [Dental patterns of the stem-group hexanchoid shark, Notidanoides muensteri (Elasmobranchii, Hexanchiformes)]. Thus, there seems to be a constraint in the evolution of hexanchiforms towards dignathic heterodonty.

We thank the reviewer for this insightful reference, which we have incorporated in our manuscript, together with a short discussion. The apparent lack of high dignathic heterodonty in stem hexanchiforms aligns with lower dignathic heterodonty in recent *Chlamydoselachus* (the sister clade), suggesting that high dignathic heterodonty in this group might be apomorphic. However, given the strong tooth morphological resemblance between *N.muensteri* and extant hexanchids, this case study presents a compelling example of heterodonty-level complexity being less conserved than tooth-level complexity, which is in line with the statistical results displayed in Figure 3, but is an exception from the overall higher deep-time conservation of dignathic than monognathic heterodonty features. Since we are assuming a mostly statistical/synthetic perspective on dental traits in this paper, we cannot dedicate an extended discussion to the group-specific evolution of dignathic heterodonty. However, we have now integrated some more explicit discussion on superorder-level differences between the modularity between mono- and dignathic heterodonty, also citing the case of *N.muensteri* (3.section of Results).

Including fossil taxa could increase our understanding of evolutionary pathways of such adaptive traits better than just using extant forms to identify possible constraints.

We agree with the reviewer, and did actually consider this option in the beginning of the project. However, a meaningful and phylogenetically consistent inclusion of fossil specimens could not be done owing to the scarcity of sufficiently complete dentitions for most shark clades. This is an actual hard constraint and one major reason why most comparative and evolutionary studies on shark dental features focus on isolated teeth.

Conversely to squalomorphs, galeomorph sharks have a more gradual (sequential) heterodonty.

Yes, this has been documented by our proxy measure of the largest shape difference between any two teeth within a jaw and their sequential heterodonty, cf. Figure 4(C) and Figure S2(B).

Mapping all information derived from the shape analyses across all hierarchical levels indicates that the various heterodonty patterns and tooth-level complexities evolved independently several times based on information derived from extant taxa. Importantly, this approach reveals that no significant positive correlation between genetic distance and heterodonty differences for genetically close or moderately distant species exist, but genetically most distant species display significantly higher heterodonty differences. This indicates that heterodonty can change relatively unconstrained and thus is highly evolvable (lines 117-118). There seems to be no strong constraints preventing closely related taxa from developing divergent heterodonty patterns (line 273-275).However, this seems odd. If only genetically distant species exhibit significantly higher heterodonties, why can heterodonty then change relatively unconstrained in low-to-moderately genetically related taxa? I would expect such a process if the pattern also is significant in closely related species.

Our results indicate that, statistically, genetic distance is generally not correlated with phenotypic distance for heterodonty traits. This means that pairs of closely related species are not less likely to show *differences* in heterodonty than pairs of more distant taxa. A notable exception is dignathic heterodonty, for which we do see significance of correlations with genetic distance for taxa that are somewhat distant (mid-range of genetic distances, cf.Figure 3). Thus, our result is, overall, consistent with absence of constraints. We assume that the reviewer refers to the “tail end” within the graph in Figure 3(A) for heterodonty differences. Therein, it can be seen that the significance of heterodonty changes between the genetically most distant clades, i.e. galeomorph-versus-squalomorph distances, tend to be higher than for mid-range distances. Upon in-detail measurement – now appended as Supplementary Figure F9 -, we find that this significance-of-correlation peak is chiefly driven by *Hexanchus* which presents one of the highest dignathic heterodonty scores, but also features the highest total genetic distances from any galeomorph clade. We had glossed over this aspect, because it reflected merely a disproportional influence of one “extreme” *genus* (given the structure of the phylogeny, this end of the distribution tends to be less reliable, since driven by fewer taxa). Note that this is not an artifact but an intrinsic limitation of the method: while the central bulk of the measurements represents a mix between diverse pairs of species, the terminal end over-represents a few most diverged taxa. However, we agree that it is important to interpret this finding explicitly. This is now done in the Results section, thereby neither omitting nor ignoring this limitation. As mentioned, we also provide another supplementary figure showing that single taxa have a disproportionate effect only on the tail end of the shown distribution, but not for the remainder of the curves.

The peak in correlation significance for dignathic heterodonty around dG=0.45 seen in Figure 3(B) can be interpreted as the transition between within-superorder to between-superorders ranges of genetic distances, and may reflect the generally higher dignathic heterodonty among squalean sharks, resulting in a significant increase of dignathic heterodonty at this point.

In a nutshell, constraints are understood to exist only in the case of strong correlation between genetic and phenotypic differences, as in that case, genetic similarity prevents phenotypic changes with respect to the trait in question. High p-values of correlation, as for most heterodonty measures between more closely related taxa, indicate the opposite, i.e. absence of constraints.

The study is based on 51 extant species, each obviously represented by a single specimen. While this rather large number of species and corresponding dental information, it does not catch the intraspecific and possible interspecific dental variations within extant taxa, let alone in fossil taxa. This information, however, could provide additional information about evolutionary pathways and constraints!

We agree with the reviewer that it would be very insightful to delve into different levels of phylogenetic depth, including intraspecific variation, which includes variation between populations, sexes and age cohorts. While this would have certainly been possible for a few taxa (such as *Scyliorhinus*, lamnids, and carcharinids), no comparable data is available for the majority of taxa. Therefore, including variation in a partial way would have created a bias and might not have yielded class-level or generalizable conclusions. However, we are considering publishing species-level variation for a few selected taxa in a separate study with a different primary focus.

In lines 121-133, the authors state some tooth-level complexity measures might serve as a moderately better proxy for relatedness than heterodonty. Which measures are these? It would be useful to list these.

We meant to refer mainly to the Fourier analysis-based complexity measure whose phylogenetic signal exceeds all dentition-level complexity measures. We have now reformulated our statement accordingly in order to be clearer.

Interestingly, including ecological features, especially depth distribution in the analyses results in a better separation of the two superorders indicating that heterodonty and tooth complexity patterns result from adaptive morphological changes. Thus, the significantly higher dignathic heterodonty found in many squalomorphs would be the result of deep-water adaptation.

This is possible, or adaptation towards food sources and hunting strategies correlated with it. We have also added a supplementary figure to visualize this relationship better.

But it remains ambiguous why this is. And moreover, dignathic heterodonty is absent in deep-water galeomorphs, such as, e.g., Apristurus. I would suggest to list such exceptions and to discuss possible reasons for presence / absence of dignathic heterodonties in relation to habitat in more detail.

This is indeed an interesting topic for speculation and it might come down to micro-niches within the deep sea and/or developmental constraints, i.e. the degree of inter-jaw modularity. We had not gone into deeper speculation about group-specific findings as the paper aimed more at a class-level, statistical, analysis, and there is not enough data to statistically explore evolutionary hypotheses regarding heterodonty trends *within* orders. However, as this is indeed a fascinating topic for speculations, we have now included a paragraph in the discussion. We suspect it might have to do with distinct, specialized, feeding mechanisms that in case of squalomorphs make use of the higher modularity between mono- and dignathic heterodonty in this clade. Overall, deep-sea cat sharks do not feature qualitatively different dentitions than species inhabiting shallower waters, indicating developmental constraints within this clade.

In lines 174-175, the authors state that dental variation evolves dynamically within sharks. This result seemingly relies on the combination of the shape analyses with phylogenetic information. It remains unclear what dynamically means. Moreover, such a statement should be supported be specific trait analyses.

With “dynamical evolution”, we referred to the finding that there is no obvious specific phylogenetic signal for most dental traits, suggesting that dental complexity has increased, and decreased, repeatedly and independently. We have rephrased the statement slightly to make this point less ambiguous.

The authors continue to compare individual tooth shape development in sharks with that in mammals, stating that Hox genes are important in differentiating tooth classes in mammals, while the role of Hox genes in sharks remains ambiguous. A major difference between both groups, however, is the presence of a dental lamina in sharks, which is important in tooth development, and which is absent in mammals. Could it that this difference is more important than Hox genes? Compare, e.g., various publications by Gareth Fraser (see https://www.fraser-lab.net/publications.html).

Certainly, the dental lamina in elasmobranchs might be connected to a higher variability of shark tooth shapes in the jaw, as it is linked to a continuous odontogenesis with exposure to the local environment (e.g. mechanical cues from the jaw and from feeding). However, mammals show discrete tooth classes linked to Hox gene expression, and our impression is that dental variability tends to be more gradual in sharks, suggesting that there might be qualitative differences between the two classes in how within-jaw tooth shapes are generated. In other words, while the DL may have a role in generating overall and possibly more gradual heterodonty in sharks, Hox genes are involved in defining discrete dental types in mammals. We have now expanded our discussion accordingly.

According to the results, habitats correlate with feeding habits representing a proxy for a species' primary food sources. Based on this, two discrete clusters representing major ecological-functional strategies are identified, i.e., shallow-water habitats, crustacean/small invertebrate diets and open-sea habitats, large preys, with a sub-cluster of deep-sea cephalopod feeders. However, I wonder if this is not an oversimplification?

This paper presents a mostly statistical perspective which forces us to make certain oversimplifications (although there is no gold standard of what constitutes the best level of lumping data into categories), and the two clusters are actually an *outcome* of our analysis (cf.Figure S13). Overall, the limiting factor here, again, is the evenness of data resolution, with some species having an excellent trophic data record, and others being described from a single caught specimen supplemented by some inference from related species. Thus, we decided to consider coarse categories to remain commensurate with the heterogeneity of data, while outlining these limitations.

Nevertheless, the results show that monognathic heterodonty is stronger correlated with ecological traits than dignathic heterodonty.The study importantly shows that evolutionary change tends to developmentally originate from alterations in higher-level cues rather than from individually tinkering with individual teeth. This indicates that developmental and ontological nestedness could be a major biological principle.Overall, the authors achieved the targeted goals based on the data set used here. But as stated above, inclusion of more specimens of each species and more species per clade as well as extinct taxa might provide additional information about inter- and intraspecific variations in heterodonty patterns through time and corresponding constraints. This also might better show, if heterodonty patterns are really highest in genetically distant taxa.

Thank you for your suggestions that will be central in future follow-up projects. In this manuscript, we have elaborated possible limitations to our study such as the inescapable heterogeneity of ecological data. In addition, as elaborated previously, we now discuss more comprehensively the salient heterodonty differences between the genetically most distant species.

Nevertheless, this study is the first one using combined methodological approaches to establish the significance of heterodonty patterns on various hierarchical scales and thus certainly will serve a roadmap for future studies into the development of teeth and dentitions in sharks, and also beyond.

Yes, we feel – as the reviewer – that there is substantial potential for future extensions. Thank you.

This is a very important study. But I have to admit that it was quite difficult to read and to follow your arguments.

We have made improvements throughout the manuscript that we believe will enhance the accessibility of our argumentation. Nevertheless, the key conclusions rely on interactions of different results, which makes repetitions and a more complex structure difficult to avoid entirely.

Therefore, I would like to make some additional comments.- Include in the abstract ALL findings (results) of your study.

Due to the compact nature of the abstract, some results, especially minor ones, might naturally become condensed or omitted. We have now added the characterization of the differences between the two superorders that was indeed missing previously.

- State clearly all the goals. It was quite difficult to really identify all goals of the study as you only mention "Here, we elucidate such macro-patterns in the light of environmental and life-history traits, applying a novel morphometric tool combination. This may critically contribute to understanding the specific relevance of hierarchical organization of functional traits across an entire vertebrate class, a significant question across ecology, palaeontology and evolution". Only by reading all the manuscript and focusing on your results it was possible to really understand what the different goals were. I also would expect some research hypothesis that you could directly address in the results and Discussion sections.

We have now expanded the introduction in order to provide a clearer presentation of our research hypotheses, making this summary less synthetic and more specific. We have also included guiding hypotheses into the final paragraph of the introduction, as well as specific questions/task within the first lines of each section presenting specific Results.

- Please use systematic names instead of G1 and G2 in the text! I had to refer to the various figures to find out the colour code for the two major groups in one figure, then refer to the next figure to correlate the colour code / systematic name with G1 and G2, respectively to understand what you meant.

G1 and G2 are no actual groups in the systematic sense, but clusters emerging from the combination of tooth-level and dentition-level traits. We now made this aspect clearer in the respective figure caption and introduced a divergent color code in order to make the differences between the two figures more salient and less confusing.

- I would strongly suggest explaining the results of the graphs shown in Figure 4C in more detail in the text. In many instances, you simply make a statement without properly discussing it.

We agree that Figure 4(C) contains a large array of individually relevant results. We think that we have already covered all significantly clade-specific findings in the last paragraph of the third section of the Results. To ensure, we have added a cross-reference to this figure, facilitating association. We have also expanded the Discussion section where these differences are contextualized.

- What makes the manuscript difficult to read is the structure and the sometimes jumping around between topics. I would strongly suggest to alter the structure and address the different research questions / hypotheses so that there is a direct link between the goals indicated and the results and discussions.

Agreed. This impression may reflect the multitude of different angles we were trying to encompass in this manuscript, their inter-connectivity which is sometimes required to arrive at our core conclusions, and our attempts to keep the manuscript as concise as possible. As suggested above, we have now added introductory hypotheses/statements/questions for all sections of the results part where we found them missing.

We have also improved sequence and connections throughout the Discussion section and ensured congruence with the results part where this was possible.

Reviewer #2 (Recommendations for the authors):Zimm et al. investigated the relationships between dental traits, including heterodonty and complexity, with various aspects of shark biology, analysing a range of data from pre-existing sources in a novel and creative way. They find that traits coded on an intermediate level e.g. within a jaw show a low phylogenetic signal and a relatively strong relationship with ecology, suggesting significant adaptability of these traits. Particularly impressive is the quantification of these dental traits. The number of methodological approaches used to measure complexity and heterodonty is remarkable, and well-reasoned. There is excellent attention paid to the context of the study, given work that has been done previously in mammals, and the phylogenetic context is given through the genetic distance approach, which is important, and not always done in similar trait analyses for other clades.The most significant aspect of the paper that could be expanded is the discussion of the implications of these results.

Thank you for this suggestion. Given the multiple research directions this paper tries to integrate, we have tried to be as concise and synthetic as possible, yet we agree that some of the directions of the discussion may deserve to be expanded. In particular, we have expanded the discussion on the use of our result for the reconstruction of marine paleo-ecosystems as well as on the specific differences between the superorders.

In the abstract, introduction and discussion, the fossil record of sharks is mentioned, and one of the papers frequently cited in the introduction (Eronen et al., 2010) describes the use of traits, including tooth shape to reconstruct environments through the application of ecometrics. This has been done in mammals, but not, as far as I'm aware, using such a range of scales as in this study. Given the attention given to potential palaeoecological applications of the results here, I would suggest there could be further discussion of this.

Thank you, we have expanded the discussion on this point, which indeed enhances the scope of this paper. We agree that finding a strategy for reconstructing past ecosystems using the dental record that can be applied to marine ecosystems is an extremely important goal and our paper might canalize those efforts.

In particular, I wonder if a discussion of the availability of tooth rows to enough of an extent for measurement of heterodonty would be useful, as well as a consideration of the potential use of community-level shark traits for an assessment of water depth, for example.

This might indeed be an interesting strategy for paleo-ecology. We have added this valuable suggestion in our discussion.

In addition to the comments in the public review, I noticed a few small details that I would suggest addressing, although in general I thought the presentation of the work was excellent.• In general, it would be helpful to restate the definitions for acronyms in all figure captions. There are a lot, understandably, but for a reader it would be helpful to have the definitions repeated in the captions e.g. for HMS, HMT, HMX, HDG.

We have now added abbreviation glossaries into the figure captions.

• Figure 1 – Species names should be italicised in the figure. Squalomorphii and Galeomorphii are italicised in the caption but not in the rest of the text.

We have reviewed naming conventions in taxonomic categories and those tend to be specific to different parts of the Tree of Life. For complex animals, we found that the use of italics is commonly restricted to categories below the family level (although we recognize that divergent opinions exist). We think that following the most common practice would be preferable.

• Introduction – final paragraph could be expanded and made more specific about the methods and rationale

We have now made our core questions more explicit at the end of the introduction.

• Page 3, line 98 – I suggest making the clause "the main outliers…" into its own separate sentence, and I wonder if it would be better to use the full name rather than "squalean" which I don't think was previously defined

We use the names squalean and galean as they are commonly, albeit perhaps less frequently or ubiquitiously, used in parts of the elasmobranch literature, which is why we think there is some value in using them. However, in order to make the paper more inclusive for readers with different backgrounds, we introduce the synonyms squalomorph/squalean and galeomorph/galean the first time they appear in the paper where we discuss differences between superorders.

• Figure 2 – This may be personal preference, but I think it would be good to edit the figure so that the tip names are italicised and separated by a space rather than an underscore e.g. Carcharhinus sealei. The alignment of the bottom part of the figure between the different components is different and would be worth editing in the software used to make the figure. Should Selachimorpha be italicised in the caption? Other clade names in the text and in the figure are not.

We realized that in this figure, using italics would make the names less readable due to the size of the figure. We have therefore made an exception to the convention for practical reasons. We have removed the underscore.

• Section 4 (corresponding to Figure 5) – the choice of colour here is quite misleading, given that the same colours (or very similar) have been used to differentiate between the Squalomorphii and Galeomorphii in previous figures. Given that these groups do not simply correspond to the two major clades of sharks considered in Figure 4 for example, I think it would be preferable to use different colours to denote G1 and G2 in Figure 5.

We intended to make the graphics somewhat consistent in color scheme, but we agree that emphasizing the differences between phylogenetic groups and phenotypic clusters by choosing different color codes prevents confusion and is to be preferred. We have therefore changed the color code.

• It would also be good to have included in the supplementary material the group assignment for each shark species included in the study. I note that this is included in the Github, but not in the spreadsheets available in the Supplementary Material.

Thank you for pointing this out; we have added a column "group" it this file: masterfile.data on Github, containing all measures used in the paper as columns.

• Line 365 – how many taxa was this substituting of close relatives used? It would be good to have an estimate for this.

This information is provided in the ecological table provided as supplementary material (Sharks_eco_refs.xlsx).

• Line 374 – how is "extreme" defined? Similarly, the "exceptional reports" of line 379

We excluded explicitly “unconfirmed” reports or outliers that were substantially larger than the bulk of reports, suggesting unreliable or exaggerated records. The rationale being that, even if those reports were true, they would be likely outside the “ecologically typical” or “representative” range of the respective species and, therefore, less suitable for a statistical study. We have added a clarifying sentence in the methods section.

• The list of ecological descriptors would be more useful if it were distributed in the text according to where the different categories are mentioned. For example, the prey categories could be listed at Line 368

Agreed, we have now moved/distributed the lists to the appropriate places.

• In the methods, you mention that the complexity is defined as the sum of the constituent measures. I'm not sure if I'm misunderstanding, but are the different measures actually comparable in this way? I.e. are the different measures on the same scale in a way that would allow them to be summed? I'm struggling to tell from the raw data available in the Github, but if not, this would be quite problematic.

Yes, we ensured consistency in two ways: (1) equal representation of weakly inter-correlated (but highly intra-correlated) orthogonal groups of measures, producing to three different groups of measures, cf. Figure S4: Cx.exec, Cx.four and Cx.ang; (2) normalization of the measures, with a commensurate range of values between 0 and 1.